# Influencing Humans to Conform to Preference Models for RLHF

## Abstract

Designing a reinforcement learning from human feedback (RLHF) algorithm to approximate a human's unobservable reward function requires assuming, implicitly or explicitly, a model of human preferences. In sequential decision making tasks, a preference model that poorly describes how humans generate preferences risks learning a poor approximation of the human's reward function. In this paper, we conduct human studies to assess whether one can influence the expression of real human preferences to more closely conform to a desired preference model. Importantly, our approach does not seek to alter the human's unobserved reward function. Rather, we change how humans use this reward function to generate preferences, such that they better match whatever preference model is assumed by a particular RLHF algorithm. We introduce three interventions: showing humans the quantities that underlie a preference model, which is normally unobservable information derived from the reward function; training people to follow a specific preference model; and modifying the preference elicitation question. All intervention types show significant effects for at least one preference model, providing practical tools to improve preference data quality and the resultant alignment of learned reward functions. Overall we establish a novel research direction in model alignment: designing interfaces and training interventions to increase human conformance with the modeling assumptions of the algorithm that will learn from their input.

## 1 Introduction

Aligning agent behavior with human preferences is a central goal of reinforcement learning from human feedback (RLHF). This process generally assumes a model of human preferences, a probability distribution over a human's rankings of pairs of trajectory segments based on their reward function. The RLHF algorithm cannot observe the human's reward function and instead must approximate it from preferences.

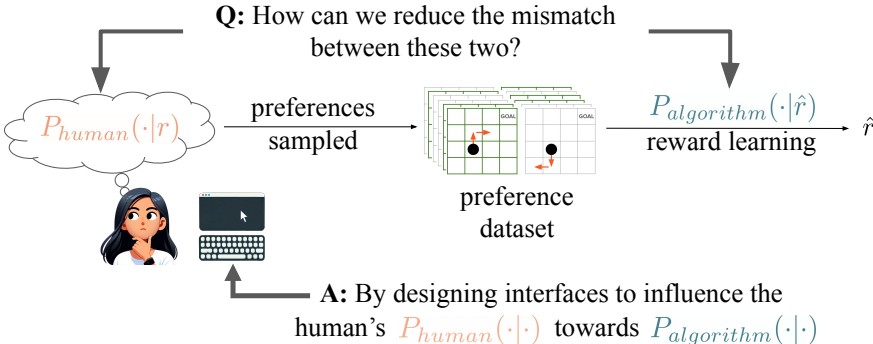

Figure 1: Our proposed method of influencing human preferences. We design interfaces to influence the human's preferences without changing their underlying reward function.

Prior work has explored different choices for this preference model and provided evidence that, in sequential decision making tasks, the more aligned the RLHF algorithm's preference model is with how humans generate preferences, the more aligned the learned reward function is (Knox et al., 2022). Most work assumes that human preferences arise probabilistically from *partial return*, the sum of rewards over a trajectory segment. By this assumption, humans presented with two

trajectory segments tend to prefer the one that accrues greater reward, as measured by their underlying latent reward function (Christiano et al., 2017; Ouyang et al., 2022). Other work has challenged whether partial return is a sufficiently descriptive model of human preferences and proposed alternative models of human preference (Knox et al., 2022; Kim et al., 2023; Marklund & Van Roy, 2024). We instead take a *prescriptive* approach to learning from human preferences. Specifically, we propose designing training and preference elicitation interfaces to influence humans to better conform to a chosen preference model. We study influence toward three preference models: the partial return preference model; the *regret* preference model, which is based on each segment's deviation from optimality (Knox et al., 2022); and the *change-in-expected-return* preference model, which is based on the expected outcomes of a segment rather than the decisions in the segments. See Figure 1 for a visual summary of our interventions.

The interventions we introduce primarily concern sequential decision making tasks, where different preference models may induce different preferences over trajectory segments. In contrast, most RLHF applications for training Large Language Models (LLMs) focus on single-step decisions, effectively collapsing differences between preference models (Knox et al., 2024). We anticipate that our approach will be particularly useful for training LLM for longer horizon decision processes.

We introduce three methods and evaluate how each method can influence humans towards these different preference models. First, in the PRIVILEGED experiment, we present subjects with calculations of regret or partial return during preference elicitation, thus providing the information needed to exactly follow the *target preference model*. This first intervention is a proof of concept, since it requires knowledge of the reward function that is untenable in real-world preference elicitation; nonetheless, it provides a useful understanding of RLHF performance in a setting where humans are given the information required to apply the prescribed preference model. In contrast, the next two methods can be deployed in practice. In the TRAINED experiment, we train humans to follow a specific preference model. In the QUESTION experiment, we change *only* the QUESTION asked during preference elicitation. The PRIVILEGED and TRAINED methods result in close conformance with either target preference model. In a follow-up experiment, we found that training subjects to follow a specific preference model and apply it in a new domain may be overly fatiguing, resulting in no conformance to one model in the new domain. The QUESTION method also increases conformance with either preference model, but significant effects are only observed when targeting particular preference models in different settings, not across all preference models. We summarize all intervention results in Table 1.

This paper's main contribution is a novel, straightforward approach to improving model alignment: training humans and designing interfaces to increase human conformance with the assumptions of the algorithm that learns from their input. Our experiments yield guidance for RLHF practitioners and lay groundwork for future research at the intersection of interface design and learning from human input, RLHF especially. The code for all computational experiments and interfaces for collecting preferences, as well as the collected datasets of human preferences, are available here.

## 2 Related Work

### 2.1 Learning reward models from human preferences

Extensive research has explored learning from human preferences for RLHF. This research includes RLHF approaches that explicitly learn a reward function (Christiano et al., 2017; Ibarz et al., 2018; Sadigh et al., 2017; Lee et al., 2021a;b; Ziegler et al., 2019; Ouyang et al., 2022; OpenAI, 2022; Bıyık et al., 2022; Wang et al., 2022; Bai et al., 2022; Glaese et al., 2022; Knox et al., 2022; Touvron et al., 2023; Ethayarajh et al., 2024) and other approaches that directly learn a policy or advantage function from human preferences (Rafailov et al., 2024; Knox et al., 2024; Hejna et al., 2023). All of the algorithms in the works cited above assume one of the models of human preference that this paper uses to design preference elicitation interventions. Our investigation of how to increase human conformance with the assumed preference model is compatible with and strengthens this past research.

Other research has sought agent alignment by developing preference models that better model human preferences, such as by assuming that human preferences arise from a segment's regret (Knox et al., 2022) or

weighted sum of non-Markovian rewards (Kim et al., 2023), instead of a segment's sum of Markovian rewards as is commonly assumed. However, improving the preference model still results in some gap between the preference model and actual human preferences, which are subject to difficult-to-model confounding factors and individual differences. Our research seeks to close this gap for whatever preference model is chosen. Further, even if we had a *perfect* descriptive model of how *all* humans generate preferences, we may want preferences to be generated by a different preference model that permits more tractable algorithms, greater sample efficiency, or some theoretical guarantees. The methods we introduce can also help in these settings.

## 2.2   The measurement of human preferences

While this paper focuses on sequential decision-making tasks, the underlying ideas about how people can be influenced to conform to different preference models through interface design and other interventions have broader implications in RLHF (and, perhaps, RLAIF (Lee et al., 2023)). In the context of LLM fine-tuning, RLHF is assumed to be an information-gathering exercise, where people see a prompt and provide their preferences over two or more responses. Within this setting, there is a strong assumption: that the interface by which we solicit preferences either does not matter or has been sufficiently well-designed as to solicit stable ground-truth preferences. However, prior work has found that annotators often adjust their preferences or annotations when they believe that their responses will shape a learning system; they will sometimes override their own judgments to instill desired norms into the learned models (S. Treiman et al., 2025; Treiman et al., 2024). Other work in RLHF has shown that annotators have substantial discretion when interpreting alignment principles (which explicitly solicit preferences as responses that are relatively more helpful, harmless, and honest), and that these judgments often vary across raters (Buyl et al., 2025). Together, these findings suggest that human judgments in RLHF are better understood as survey measurements rather than observations of stable underlying preferences; hence, preference modeling is an act of survey design (D'Alonzo et al., 2026).

This survey design perspective has motivated recent work that treats RLHF pipelines as objects of empirical and normative analysis. Studies of RLHF datasets have found that the values encoded in preference data are unevenly distributed across domains (Obi et al., 2024), while analyses of preference aggregation mechanisms have revealed potential instability and fairness concerns when combining divergent judgments (Feffer et al., 2023). Other work attempts to infer the implicit normative principles encoded in preference datasets by recovering the rules that best explain observed annotation patterns (Findeis et al., 2024). Collectively, this literature emphasizes that alignment procedures embed structured assumptions about how human judgments are interpreted and aggregated.

Our work complements these findings by studying the related-but-distinct setting of sequential decision-making, where preferences are expressed over trajectories rather than individual responses. In RLHF terms, this would require multi-turn interactions, which to our knowledge have not yet been widely integrated into training pipelines. We show that the expression of preferences over trajectories can be systematically influenced through interface design and training interventions, even when the underlying reward signal is held fixed. Broadly, our findings reinforce the view that human feedback reflects both latent human preferences as well as the interfaces that mediate preference collection.

## 3   Preliminaries: Preference Model Choices

When influencing human preferences, we analyze three preference models: the partial return, regret, and change-in-expected-return preference models. In this section, we explain the assumptions encoded in each preference model and how each can be used in RLHF.

Consider a Markov decision process (MDP) that represents the task environment using a tuple $(S, A, T, \gamma, D_0, r)$. $S$ and $A$ are the sets of possible states and actions, respectively. $T : S \times A \to p(\cdot|s, a)$ is a transition function; $\gamma$ is the discount factor; and $D_0$ is the distribution of start states. Unless stated otherwise, we assume tasks are undiscounted ($\gamma = 1$) and have terminal states, after which only 0 reward can be received. $r$ is a reward function, $r : S \times A \times S \to \mathbb{R}$, where $r_t$ at time $t$ is a function of $s_t$, $a_t$, and $s_{t+1}$. An MDP\r is an MDP without a reward function.

We say that an MDP has *deterministic transition dynamics* if, for all $s \in S$ and $a \in A$, the transition distribution $p(\cdot \mid s, a)$ is a point mass on a single next state $s' \in S$; that is, $p(s' \mid s, a) = 1$ for some $s'$. Conversely, an MDP has *stochastic transition dynamics* if there exists at least one state–action pair $(s, a)$ for which $p(\cdot \mid s, a)$ assigns non-zero probability to more than one possible next states.

Let $r$ refer to the ground-truth reward function for some MDP, $\hat{r}$ refer to a learned approximation of $r$, and $\tilde{r}$ refer to any reward function (including $r$ or $\hat{r}$). A policy ($\pi : S \times A \to [0, 1]$) specifies the probability of an action given a state. $Q_{\tilde{r}}^{\pi}$ and $V_{\tilde{r}}^{\pi}$ refer respectively to the state-action value function and state value function for a policy, $\pi$, under $\tilde{r}$, and are defined as follows: $V_{\tilde{r}}^{\pi}(s) \triangleq \mathbb{E}_{\pi}[\sum_{t=0}^{\infty} \tilde{r}(s_t, a_t, s_{t+1}) | s_0 = s]$ and $Q_{\tilde{r}}^{\pi}(s, a) \triangleq \mathbb{E}[\tilde{r}(s, a, s') + V_{\tilde{r}}^{\pi}(s')]$.

An optimal policy $\pi_{\tilde{r}}^*$ is any policy where $V_{\tilde{r}}^{\pi_{\tilde{r}}^*}(s) \geq V_{\tilde{r}}^{\pi}(s)$ at every state $s$ for every policy $\pi$. We write shorthand for $Q_{\tilde{r}}^{\pi_{\tilde{r}}^*}$ and $V_{\tilde{r}}^{\pi_{\tilde{r}}^*}$ as $Q_{\tilde{r}}^*$ and $V_{\tilde{r}}^*$, respectively.

### 3.1 Reward Learning from Pairwise Preferences

RLHF typically learns a reward function by minimizing the cross-entropy loss—i.e., maximizing the likelihood—of observed human preference labels (Christiano et al., 2017; Ibarz et al., 2018; Wang et al., 2022; Bıyık et al., 2021; Sadigh et al., 2017; Lee et al., 2021a;b; Ziegler et al., 2019; Ouyang et al., 2022; Bai et al., 2022; Glaese et al., 2022; OpenAI, 2022; Touvron et al., 2023; Hejna III & Sadigh, 2023). Here, we examine the preliminaries needed to precisely define and this loss function when learning from preferences. **Segments** Let $\sigma$ denote a segment starting at state $s_0^{\sigma}$. Its length $|\sigma|$ is the number of transitions within the segment. A segment includes $|\sigma| + 1$ states and $|\sigma|$ actions: $(s_0^{\sigma}, a_0^{\sigma}, s_1^{\sigma}, a_1^{\sigma}, ..., s_{|\sigma|}^{\sigma})$. In this problem setting, segments lack any reward information. As shorthand, we define $\sigma_t \triangleq (s_t^{\sigma}, a_t^{\sigma}, s_{t+1}^{\sigma})$. Additionally, we denote the *partial return* of a segment $\sigma$ as $\Sigma_{\sigma} \tilde{r}$ for some $\tilde{r}$, where $\tilde{r}_t^{\sigma} \triangleq \tilde{r}(s_t^{\sigma}, a_t^{\sigma}, s_{t+1}^{\sigma})$ and $\Sigma_{\sigma} \tilde{r} \triangleq \sum_{t=0}^{|\sigma|-1} \tilde{r}_t^{\sigma}$.

**Preference datasets** We denote a preference dataset as $D_{\succ}$, which comprises samples of preferences over pairs of segments. Each sample is represented as $(\sigma_1, \sigma_2, \mu)$. Vector $\mu = \langle \mu_1, \mu_2 \rangle$ represents the preference; specifically, if $\sigma_1$ is preferred over $\sigma_2$, denoted $\sigma_1 \succ \sigma_2$, $\mu = \langle 1, 0 \rangle$. $\mu$ is $\langle 0, 1 \rangle$ if $\sigma_1 \prec \sigma_2$ and is $\langle 0.5, 0.5 \rangle$ for $\sigma_1 \sim \sigma_2$ (no preference). For a sample $(\sigma_1, \sigma_2, \mu)$, we assume that the two segments have equal lengths (i.e., $|\sigma_1| = |\sigma_2|$) and the same start state (i.e., $s_0^{\sigma_1} = s_0^{\sigma_2}$).

**Loss function** When learning a reward function from a preference dataset, $D_{\succ}$, preference labels are typically assumed to be generated by a preference model $P$ based on an unobservable *ground-truth* reward function $r$. We learn $\hat{r}$, an approximation of $r$, by minimizing this cross-entropy loss:

$$loss(\hat{r}, D_{\succ}) = -\sum_{(\sigma_1, \sigma_2, \mu) \in D_{\succ}} \mu_1 \log P(\sigma_1 \succ \sigma_2 | \hat{r}) + \mu_2 \log P(\sigma_1 \prec \sigma_2 | \hat{r}) \tag{1}$$

If $\sigma_1 \succ \sigma_2$, the sample's likelihood is $P(\sigma_1 \succ \sigma_2 | \hat{r})$ and its loss is therefore $-log P(\sigma_1 \succ \sigma_2 | \hat{r})$. If $\sigma_1 \prec \sigma_2$, its likelihood is $1 - P(\sigma_1 \succ \sigma_2 | \hat{r})$. This loss is under-specified until the preference model $P(\sigma_1 \succ \sigma_2 | \hat{r})$ is defined. Learning approximations of $r$ from preferences can be summarized as "minimize Equation 1".

**Preference models** A preference model determines the likelihood of one trajectory segment being preferred over another, $P(\sigma_1 \succ \sigma_2 | \tilde{r})$.

### 3.2 Preference Models: Partial Return, Regret, and Change-In-Expected-Return

Here we describe three preference models, two of which are commonly used when learning from human preferences. Figure 2a illustrates an example of how these models compare in an environment with deterministic transition dynamics and Figure 2b illustrates how the regret and change-in-expected-return models differ in an environment with stochastic transition dynamics. In Section 5 we detail our proposed training procedures and preference elicitation interfaces that influence human preferences to conform to a choice of preference model. We focus on three preference models that differ solely in the segment statistic they use to evaluate the desirability of a trajectory segment. We do not address other aspects of modeling human preferences, such as different assumptions about human irrationality, risk aversion, or uncertainty. However, the interventions

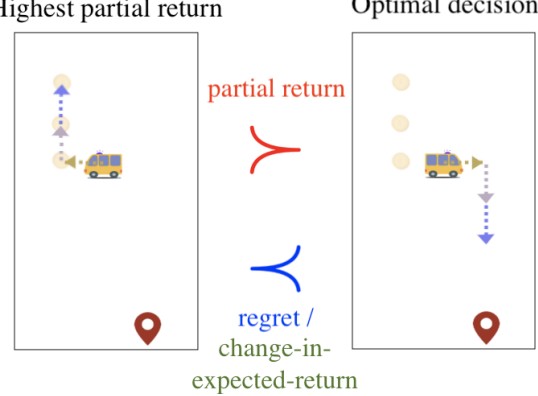

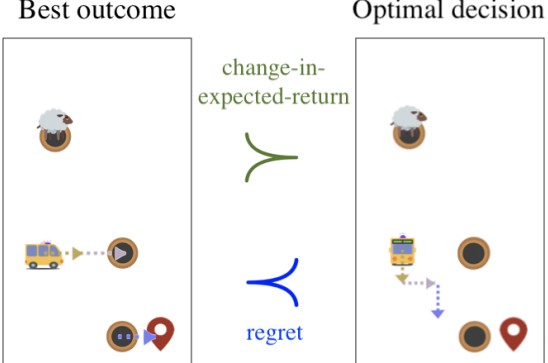

(a) On each step, the vehicle receives reward as the sum of reward components: $-1$ for every time it moves; $+1$ for collecting a coin; and $+50$ for reaching the red goal marker, which ends the episode. The partial return preference model favors the left trajectory, which ends with the highest sum of rewards, while the regret and change-in-expected-return preference models favor the right, which exhibits optimal decision-making.

(b) On each step, the vehicle receives reward as the sum of reward components: $-1$ for every time it moves; $+1$ for collecting a coin; and $+50$ for reaching the red goal marker, which ends the episode. When the vehicle enters a tunnel (brown circle), it exits from a uniformly sampled tunnel at a different location. If the vehicle exits from the tunnel with the sheep on top of it, it receives a reward of $-50$ which ends the episode. The change-in-expected-return preference model favors the left trajectory, which ends in the best outcome, while the regret and change-in-expected-return preference models favor the right, which exhibits optimal decision-making.

Figure 2: Examples illustrating disagreement between preference models in deterministic (left) and stochastic (right) environments.

proposed in this paper, which aim to influence humans towards a chosen preference model, may be generalizable to influencing humans to conform to preference models that incorporate these additional factors.

**Partial return**     The most common preference model (e.g., Christiano et al. (2017)) posits that human preferences are determined by the partial returns of the two segments, such as by a Boltzmann distribution over the partial returns of the two segments:

$$P_{\Sigma_r}(\sigma_1 \succ \sigma_2 | \tilde{r}) = logistic\Big(\Sigma_{\sigma_1}\tilde{r} - \Sigma_{\sigma_2}\tilde{r}\Big). \tag{2}$$

**Regret**     An alternative model of human preferences is based on regret (Knox et al., 2022). This model suggests that human preferences arise from the deviations of each segment from optimal decision-making, characterized by the regret of each transition within the segment. For a single transition, $regret_s(\sigma_t | \tilde{r}) \triangleq V_{\tilde{r}}^*(s_t) - Q_{\tilde{r}}^*(s_t, a_t)$. For a full segment,

$$
\begin{aligned}
regret(\sigma | \tilde{r}) &\triangleq \sum_{t=0}^{|\sigma|-1} regret(\sigma_t | \tilde{r}) \\
&= \sum_{t=0}^{|\sigma|-1} V_{\tilde{r}}^*(s_t) - Q_{\tilde{r}}^*(s_t, a_t),
\end{aligned}
\tag{3}
$$

For MDPs with only deterministic transitions, both formulations of regret are equivalent: $regret_d(\sigma_t | \tilde{r}) = regret_s(\sigma_t | \tilde{r})$. A version of the regret preference model is the Boltzmann distribution over the *negated* regret:

$$P_{regret}(\sigma_1 \succ \sigma_2 | \tilde{r}) = logistic\Big(regret(\sigma_2 | \tilde{r}) - regret(\sigma_1 | \tilde{r})\Big).$$

See Appendix B for further explanation of regret, partial return, and how these preference models differ.

**Change-in-expected-return**     We additionally consider the change-in-expected return preference model. The change-in-expected-return of a trajectory segment is defined below:

$$\Delta\text{-expected-return}(\sigma|\tilde{r}) \triangleq (\Sigma_\sigma \tilde{r} + V_{\tilde{r}}^*(s_{|\sigma|}^\sigma)) - V_{\tilde{r}}^*(s_0^\sigma), \tag{4}$$

The change-in-expected-return preference model is therefore given as:

$$P_{\Delta\text{-expected-return}}(\sigma_1 \succ \sigma_2|\tilde{r}) = logistic\Big(\Delta\text{-expected-return}(\sigma_1|\tilde{r}) - \Delta\text{-expected-return}(\sigma_2|\tilde{r})\Big).$$

For MDPs with deterministic transition dynamics, the change-in-expected-return of a trajectory segment is equivalent to its regret (Knox et al., 2022). For MDPs with stochastic transitions, however, change-in-expected-return differs from regret. The change-in-expected-return preference model accounts for the return accrued in a trajectory segment as well as the start and end state values, rather than solely accounting for a segment's deviation from optimal decision-making. The change-in-expected-return preference model is closely related to the bootstrapped return preference model introduced by Marklund & Van Roy (2024); the former assumes that an optimal policy is used to compute the start and end state value functions in an MDP with transition dynamics known to the human preference labeler, while the latter does not. We leave a consideration of the bootstrapped return preference model to future work.

**Relationship between preference models**     We refer to the partial return, regret, and change-in-expected-return of a segment as types of *segment statistics*. Knox et al. (2022) demonstrated that the regret and change-in-expected-return preference models better fits a dataset of human preferences, leading to more human-aligned learned reward functions.[1] From the regret preference model, Hejna et al. (2023) derive constrastive preference learning (CPL), which learns a policy *directly* from human preferences and thereby avoids the challenge during learning of modeling optimal value functions for $\hat{r}$ each time it is updated. Nevertheless, using the partial return preference model remains appealing given the greater simplicity of learning a reward function with it and the amount of research that has been built upon it. Knox et al. (2024) showed that the predominant method for fine-tuning large language models by RLHF can be derived from either the regret or partial return preference model, which further supports the practicality of both models, although the derivation from the regret preference model appears preferable in that it avoids arbitrarily setting the discount factor $\gamma = 0$. Therefore our goal is to assess if it is possible to influence humans to adopt a given preference model, whether partial return, regret, or change-in-expected-return. Most of our experiments are conducted in MDPs with deterministic transition dynamics. In this regime, the regret and change-in-expected-return preference models are equivalent, and

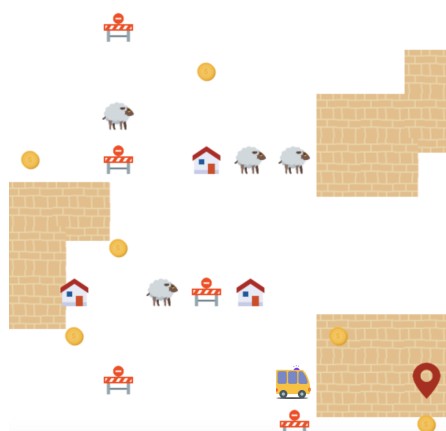

Figure 3: The delivery task shown to human subjects for gathering preferences. The yellow vehicle is the agent, and its objective is to maximize its score. Score maximization requires reaching the red inverted teardrop.

so we refer to these models as regret for simplicity. Accordingly, in such experiments in deterministic MDPs, we focus on influencing human subjects toward either the regret or the partial-return preference model, but these experiments can equivalently be viewed as influencing human subjects toward either the change-in-expected-return or the partial-return preference models. In the descriptions above of the three preference models, all are defined narrowly to assume that humans follow the Boltzmann distribution. This assumption is ubiquitous in RLHF but has limitations, which Zhu et al. (2024) overview. All three segment statistics considered, however, inform probability distributions other than the Bolzmann distribution. Therefore, in our evaluations (Section 5) we additionally test whether humans conform to the *noiseless* version of a preference model, which deterministically prefers the segment with the more desirable statistic (e.g., higher partial

---

[1]In their work, these preference models are evaluated in settings with deterministic transition dynamics, where they are equivalent. In this work, we propose treating these two preference models separately.

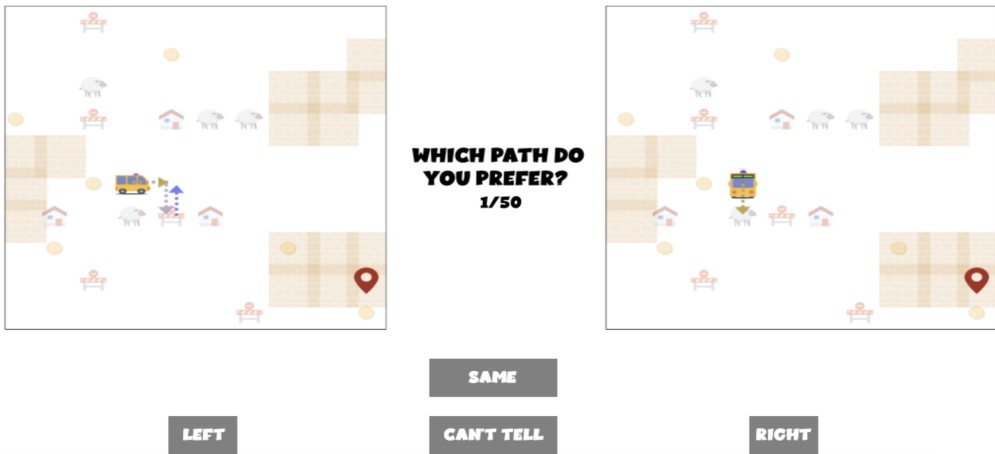

Figure 4: The baseline preference elicitation interface shown to humans annotators. All three of our experiments involve changes to this interface, whether by adding additional metrics (PRIVILEGED experiment), training the human before eliciting preferences (TRAINED experiment), or changing the elicitation question (QUESTION experiment).

return for the noiseless partial return preference model or lower regret for the noiseless regret preference model).

## 4 Experimental task and preference elicitation procedure

To empirically investigate our methodology for influencing human preferences, we collected preference datasets labeled by human subjects with IRB approval. This section provides an overview of the user interface elements shared by each experiment introduced in Section 5. See Appendix C for further details.

**Subject training** Subjects first learn about the general grid-world delivery domain, for which task instantiations are created by a map of objects and terrain. The subjects specifically are taught about the reward function (i.e., the reward values associated with each type of object or terrain) as well as about how objects and terrain affect the agent's transitions. As part of this teaching, subjects control the agent on domain maps designed to teach one or two concepts at a time. One such map—the one used to collect preferences—is shown in Figure 3. The transition dynamics for the grid-world delivery domain are always deterministic unless otherwise stated.

**Preference elicitation interface** After teaching subjects to understand the domain and delivery task, we elicit their preferences. Figure 4 illustrates a baseline version of the preference elicitation interface. In this work, we exclude preferences labeled "can't tell." After preference elicitation, a survey tests the subject's task comprehension and attentiveness, as detailed in Appendix C.4.

## 5 Experimental evaluation of three methods of influence

Aiming to decrease the gap between an RLHF algorithm's assumed preference model and how a preference dataset is actually generated by humans, we conduct random-assignment experiments for collecting human preferences. Each experiment represents a type of intervention and has three conditions that each result in a preference dataset: a control condition, an intervention condition that influences subjects to follow the regret preference model ($P_{regret}$), and an intervention condition that influences subjects to follow the partial return preference model ($P_{\Sigma_r}$). Thus, the ability of each type of intervention to influence the human towards a *target preference model* is tested with two different target preference models. When the regret and change-in-expected return preference models are not equivalent, i.e., when the environment has stochastic transition dynamics, we additionally explore influencing people to follow the change-in-expected-return preference model ($P_{\Delta\text{-expected-return}}$).

- **PRIVILEGED experiment** (Section 5.1) - The intervention of this experiment is to display information about each segment's regret or partial return under the ground-truth reward function during preference elicitation. We refer to this information as privileged because it relies upon the ground-truth reward function, which in practical settings is unknown by the code underlying the preference elicitation interface.

- **TRAINED experiment** (Section 5.2) - The intervention of this experiment is to train subjects to give preferences based upon either partial return or regret. This training procedure includes teaching people to calculate a segment's regret or partial return.

  - **TRAINED-DIFF-DOMAIN experiment** (Section 5.2.1 Evaluates this intervention when training people in one domain and eliciting their preferences in another.

- **QUESTION experiment** (Section 5.3) - The intervention of this experiment changes the question during preference elicitation (see Figure 4, "Which path do you prefer?") to one designed to encourage adherence to one of the preference models.

  - **QUESTION-STOCHASTIC-MDP experiment** (Section 5.3.1) evaluates this intervention in a domain with stochastic transition dynamics.

In all experiments, the ground-truth reward function is taught to subjects before preference elicitation to enable our analysis, but it is unavailable to the reward learning algorithm, which relies solely on the generated preference dataset. Only the PRIVILEGED experiment leverages the ground-truth reward function in its preference elicitation interface design.

## 5.1 PRIVILEGED experiment

We first study whether providing subjects with privileged information about a preference model during preference elicitation influences their preferences towards that model. Although not reminiscent of real-world data collection, presenting this privileged information probes the extent to which human preferences are susceptible to influence.

When presented with two segments for preference labeling, subjects in all three conditions are asked "Which shows better behavior?"[2] Each subject labeled preferences between 35 to 50 segment pairs. After data filtering (see Appendix C), our datasets come from from 39 subjects in the $P_{\Sigma_r}$-PRIVILEGED condition (video walk-through of the interface), 43 subjects in the $P_{regret}$-PRIVILEGED condition (video walk-through), and 42 subjects in the PRIVILEGED-Control condition (video walk-through). We refer to each condition's resultant preference dataset by the condition's name.

**Intervention details: subject training**     In the $P_{\Sigma_r}$-PRIVILEGED condition, subjects are shown each segment's partial return during preference elicitation. Likewise, subjects in the $P_{regret}$-PRIVILEGED condition are shown each segment's regret. Examples can be found in Figure 22 of the Appendix. Note that we do not explicitly instruct subjects to use the displayed segment statistics when labeling preferences. Rather, it is made visible. In the third condition, PRIVILEGED-Control, subjects are not shown any information during preference elicitation other than the visualization of the two segments that all subjects in these experiments see. See Appendix D for further details. Conditions differ only as discussed above.

**Hypothesis 1:** Presenting each segment's statistic—whether partial return or regret—will influence the human to give preferences according to this statistic.

To test this hypothesis, we compare how well the target preference model predicts the resultant dataset of preferences for the corresponding condition and for the control condition. Specifically, the preference model is given the ground-truth reward function, scaled by a constant positive factor. Given a target preference model, we compare the mean cross entropy (i.e., negative log likelihood) of each condition's preference dataset. Noting that all positive scalings of the reward function order policies equivalently and also only affect the Boltzmann distribution as a temperature parameter would, we choose the scaling constant that results in the lowest cross entropy during a grid search.

---

[2]This question was later refined for the other experiments to the baseline question in Figure 4: "Which path do you prefer?"

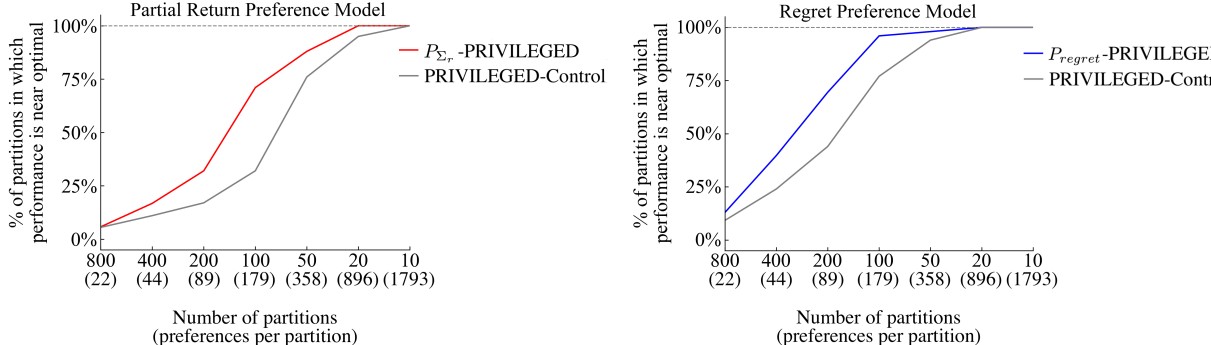

Figure 5: Learning a reward function with the partial return preference model (Left) and regret preference model (Right) from the preferences collected in the PRIVILEGED experiment. Each preference dataset is randomly partitioned into equal-sized training sets 10 times, using 10 different random seeds to control partitioning. We learn a reward function using the given preference model for each training set, and plot the percent of partitions—or equivalently training sets—in which the learned reward function induces near-optimal behavior. Note this percentage is across all partitions across all 10 seeds. To visually test for Hypothesis 2, observe the gap between the colored (i.e., red and blue) lines and the gray line.

The cross entropy losses plotted in Figure 6 support Hypothesis 1. In particular, for both intervention conditions, presenting PRIVILEGED information about the target preference model results in a dataset with a higher likelihood than that of the PRIVILEGED-Control dataset. Because samples in each condition are unpaired, we perform a Mann-Whitney U test comparing the likelihood of samples in the PRIVILEGED-Control dataset to those in the intervention condition's dataset. Both interventions result in a statistically significant effect ($p < 0.01$). Appendix I provides plots of the cross-entropy loss for all reward scaling parameters for each experiment, and more details on the statistical test.

A concern may arise that our proposed interventions merely teach human subjects more about the ground-truth reward function rather than about the target preference model. However, as we discuss in Appendix I, our full empirical results illustrate that this concern is unwarranted.

In a different analysis, we remove the assumption that humans give preferences according to the Boltzmann distribution—used in the above likelihood test—by computing the accuracy for a noiseless version of each preference model. A Fisher's exact test also finds significant effects for both interventions ($p < 0.01$), supporting Hypothesis 1. See Appendix K for details about our accuracy testing.

**Hypothesis 2:** Presenting each segment's statistic leads to learning more aligned reward functions with the preference model corresponding to this statistic.

For each condition's dataset, we learn a reward function $\hat{r}$ by minimizing Equation 1 when assuming $P_{regret}$ or $P_{\Sigma_r}$. In all experiments, regret is approximated via the algorithm proposed by Knox et al. (2022), employing successor features to efficiently compute differentiable estimates of the value functions in Equation 3. Value iteration (Sutton & Barto, 2018) then computes the approximately optimal $Q_{\hat{r}}^*$ function, and we then derive the maximum-entropy optimal policy from $Q_{\hat{r}}^*$, which uniformly randomly selects among all optimal actions. We assess the mean return of the resulting policy relative to the ground-truth reward function $r$ over the initial state distribution $D_0$.

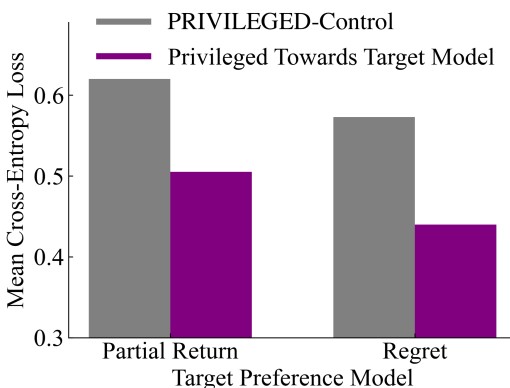

Figure 6: For the PRIVILEGED experiment, the mean cross-entropy loss over each condition's preference dataset with respect to the target preference model. If the loss is lower for an intervention's dataset than for the PRIVILEGED-Control dataset, then the former is better predicted by the target preference model. Performing a Mann-Whitney U test results in $p < 0.01$ for both conditions.

We normalize the mean return of a policy $\pi$, $V_r^\pi$, using the formula $(V_r^\pi - V_r^U)/(V_r^* - V_r^U)$. Here, $V_r^*$ denotes the expected return of an optimal policy and $V_r^U$ denotes the expected return of a uniformly random policy, both computed over $D_0$. A normalized mean return greater than 0 is better than $V_r^U$, while a value of 1 corresponds to optimal performance. We consider a normalized return above 0.9 to signify near-optimal performance.

To test performance when learning from different preference dataset sizes, for each dataset, we randomly assign human preferences to different numbers of same-sized partitions. Each partition is used as a training set, and we evaluate with 10 different random seeds that control how preferences are partitioned. Results are shown in Figure 5, with reward learning details outlined in Appendix L for all experiments. Appendix L additionally presents results showing the percentage of partitions in which better than uniformly-random performance was achieved and results when learning a reward function with both preference models for all datasets.

Figure 5 supports Hypothesis 2. Learning with the partial return preference model from the $P_{\Sigma_r}$-PRIVILEGED dataset results in reward functions that induce near-optimal behavior more often than when learning from the PRIVILEGED-Control dataset for all partition sizes, except for the largest partition size where performance is matched. A comparable pattern is observed when learning from the $P_{regret}$-PRIVILEGED dataset with the regret preference model. These results indicate that influencing human preferences towards a chosen preference model can increase the alignment of the reward function learned from those preferences.

## 5.2 TRAINED Experiment

In real-world practice, RLHF labelers do not have access to the ground-truth reward function during preference elicitation. Therefore, in this and all remaining experiments, we focus on interventions to improve preference model alignment that are feasible in practice. In the TRAINED experiment, we evaluate training humans to follow either the partial return or regret preference models during preference elicitation. All experimental details are the same as those of the PRIVILEGED experiment unless otherwise noted.

This experiment consists of three conditions. In the $P_{\Sigma_r}$-TRAINED and $P_{regret}$-TRAINED intervention conditions, subjects are taught to follow the corresponding preference model. In the TRAINED-Control condition, subjects are not taught about any specific preference model and are trained identically as all subjects in the PRIVILEGED experiment. The conditions differ only by how subjects are trained. Subjects are randomly assigned to a condition until each condition has data from 10 subjects. Unlike in the PRIVILEGED experiment, if a subjects' preference data is removed because of poor task comprehension or inattentiveness (see Appendix C.4), their slot is re-opened for random assignment, including their exact set of segment pairs to be labeled with preferences. With this replacement strategy, we gather preferences between the same set of segment pairs for each condition, which produces paired data. We discuss this design choice in Appendix I. A video walk-through of the interface used for the $P_{\Sigma_r}$-TRAINED condition is available here, for the $P_{regret}$-TRAINED condition here, and for the TRAINED-Control condition here.

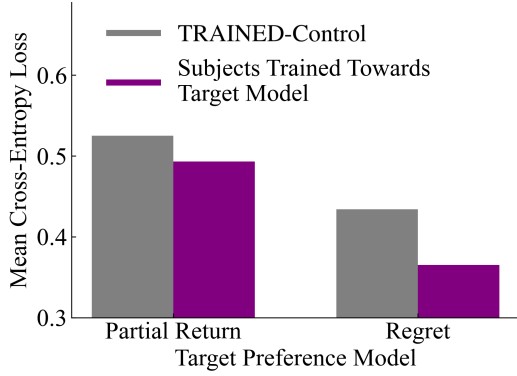

Figure 7: For the TRAINED experiment, mean cross-entropy loss over each condition's preference dataset with respect to the target preference model. Lower is better. Performing a Wilcoxon paired signed-rank test results in $p < 0.01$ for both conditions.

**Intervention details: subject training**     During training, subjects in the intervention conditions learn about the domain's dynamics and reward function, as well as the segment statistic specific to their condition (partial return or regret). They are shown the segment statistic while interacting with the delivery domain and taught how to compute it. We then provide subjects with a detailed example of how to use the taught segment statistic to generate preferences, have them practice with feedback, and finally ask them to label

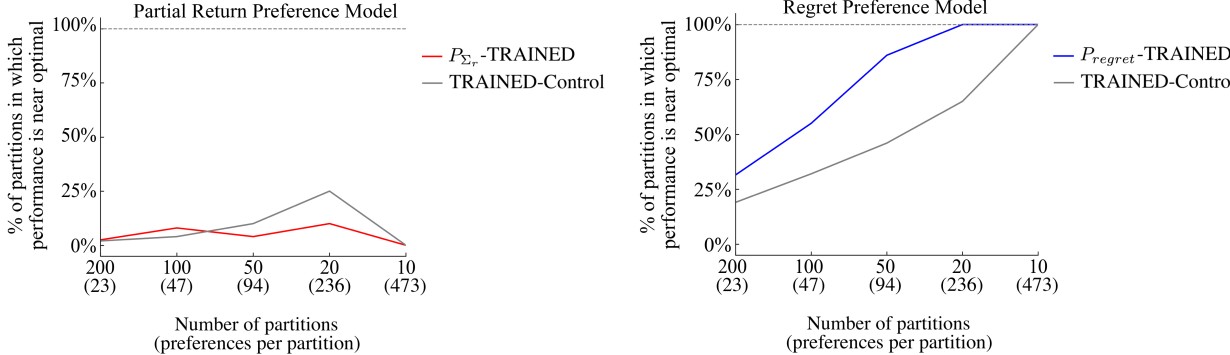

Figure 8: Learning a reward function with the partial return preference model (Left) and regret preference model (Right) from the preferences collected in the TRAINED experiment. See Figure 5 for more details on how this figure was generated.

preferences over segment pairs from the delivery task. Subjects label preferences for 50 segment pairs with the elicitation question changed based on the condition.

For this experiment, the grid-world domain used for both subject training and preference elicitation has deterministic transition dynamics. In deterministic MDPs, the regret segment statistic (Eq. 3) and the change-in-expected-return statistic (Eq. 4) are equivalent. Therefore, in the $P_{regret}$-TRAINED condition, we instruct subjects to compute regret using the change-in-expected-return formulation. This allows us to teach a conceptually simpler quantity while remaining consistent with the regret model in this environment. Further details on the training protocol are provided in Appendix E.

**Hypothesis 1:** Training a human to follow a specific preference model will influence the human to give preferences that are more in conformance with that model.

We follow the procedure outlined in Section 5.1 to evaluate evaluate Hypothesis 1 with results shown in Figure 7. Hypothesis 1 is supported; the intervention condition datasets are more likely under the respective target preference model than the control condition. Because samples are paired, we perform a Wilcoxon paired signed-rank test comparing the likelihood of samples in the control dataset to those in the dataset from training humans to follow a preference model, finding a statistically significant difference at $p < 0.01$ that further supports Hypothesis 1. See Appendix I for details. Additionally, we find that the accuracy over both the $P_{\Sigma_r}$-TRAINED and $P_{regret}$-TRAINED datasets is notably higher than the accuracy over the TRAINED-Control dataset given the noiseless version of the respective target preference model. These results, detailed in Appendix K, additionally support Hypothesis 1 with statistical significance at $p < 0.01$.

**Hypothesis 2:** Training humans to follow a specific preference model leads to learning more aligned reward functions with that preference model.

Figure 8 illustrates partial support for Hypothesis 2. Learning from the $P_{regret}$-TRAINED dataset results in reward functions that induce near-optimal performance more often than learning from the TRAINED-Control dataset when using the regret preference model for learning. But, learning from the $P_{\Sigma_r}$-TRAINED dataset when using the partial return preference model leads to comparatively poor performance. See Appendix L.6 for a discussion on this result.

### 5.2.1 TRAINED-DIFF-DOMAIN: training subjects in one domain and eliciting their preferences in another

In the TRAINED experiment, we train subjects to follow a preference model and then elicit their preferences in the same domain, i.e., with the same ground-truth reward function, state space, and action space. To evaluate that intervention in a more practical setting, where the ground-truth reward function or the target domain of interest is not available during subject-training, we introduce the TRAINED-DIFF-DOMAIN experiment. In this experiment we teach subjects about the chosen preference model in the same grid-world delivery domain used for the TRAINED experiment, but then elicit their preferences in a different grid-world domain that

is visually distinct and has a different ground-truth reward function. An example of a task from the new domain is shown in Figure 9, and the new ground-truth reward function and other aspects of this domain are described in Appendix F. Unless otherwise stated, all experimental details match the TRAINED experiment.

The TRAINED-DIFF-DOMAIN experiment exactly follows the protocol of the TRAINED experiment, except that after teaching subjects about the chosen preference model using the same subject training content, we introduce subjects to the new domain. We teach subjects about the ground-truth reward function and transition dynamics for this new domain, and then elicit their preferences over trajectory pairs sampled from the new domain. In the TRAINED-DIFF-DOMAIN experiment, subjects are assigned to either the $P_{regret}$-TRAINED-DIFF-DOMAIN, $P_{\Sigma_r}$-TRAINED-DIFF-DOMAIN, or TRAINED-DIFF-DOMAIN-Control conditions, mirroring the three intervention conditions from the TRAINED experiment.

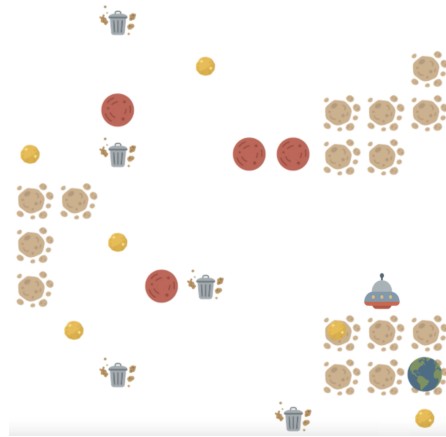

Figure 9: The task shown to human subjects for gathering preferences in the TRAINED-DIFF-DOMAIN experiment, after subjects are taught to follow a preference model in a different domain. The space ship is the agent, and its objective is to maximize its score. Score maximization requires reaching the earth.

**Hypothesis 1:** Training a human to follow a specific preference model in one domain will influence the human to give preferences are more in conformance with that model in another domain.

Results are shown in Figure 10. Hypothesis 1 is only partially supported. The loss for the dataset where participants are influenced towards the partial return preference model is lower than for the control condition's dataset under the partial return preference model, but the loss for the dataset where participants are influenced towards the regret preference model is slightly greater than for the control condition's dataset under the regret preference model. A Wilcoxon paired signed-rank test reveals a statistically significant difference at $p < 0.01$ between the likelihood of the $P_{\Sigma_r}$-TRAINED-DIFF-DOMAIN and TRAINED-DIFF-DOMAIN-Control datasets under the partial return preference model, and no statistically significant difference between the likelihood of the $P_{regret}$-TRAINED-DIFF-DOMAIN and TRAINED-DIFF-DOMAIN-Control datasets under the regret preference model. These results indicate that training subjects to follow the partial-return preference model significantly shifted their preferences toward that model in a downstream domain, whereas training subjects to follow the regret preference model did not produce a corresponding shift toward the regret preference model in the downstream domain.

We suspect that subjects' preferences were not influenced towards the regret preference model in this experiment because learning to follow the regret preference model, and then learning about a new domain, imposed a high cognitive load that fatigued participants. In contrast, the partial return preference model is comparatively easier for humans to compute. Supporting this hypothesis, subject's self reported ability to focus throughout the study and ability to follow the chosen preference model were significantly better in the $P_{\Sigma_r}$-TRAINED-DIFF-DOMAIN condition than in the $P_{regret}$-TRAINED-DIFF-DOMAIN condition. We provide further results to support this explanation in Appendix F.3. Additionally, although the regret preference model may be more difficult to explicitly follow, subjects' preferences still more closely conform to the regret preference model in the control condition than to the partial return preference model after the proposed intervention. This suggests that subjects' preferences are already more aligned with regret without any intervention, leaving less room for the intervention to further shift preferences toward regret.

While we trained subjects to compute regret for a trajectory segment, which may require excessive cognitive load, an alternative approach is to train subjects to follow the regret preference model based on intuition instead. Following the regret preference model need not be laborious, as evidenced by subjects' natural tendency to follow it in our control conditions and by the findings of Knox et al. (2022). Encouraging subjects to approximate regret through fast, intuitive reasoning may yield greater influence than requiring explicit, deliberative computation. These two approaches correspond respectively to Kahneman's System 1 and System 2 (Kahneman, 2011). We leave this direction to future work.

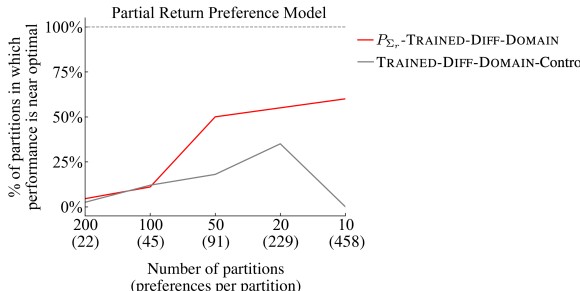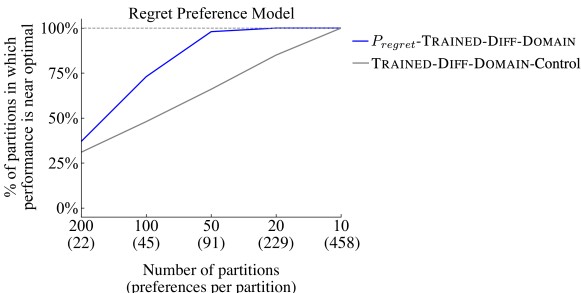

Figure 11: Learning a reward function with the partial return preference model (Left) and regret preference model (Right) from the preferences collected in the TRAINED-DIFF-DOMAIN experiment. See Figure 5 for more details on how this figure was generated.

**Hypothesis 2:** Training humans to follow a specific preference model leads to learning more aligned reward functions with that preference model in a different domain.

Figure 11 illustrates support for Hypothesis 2. Using the target preference model to learn from the dataset of preferences where humans were trained to follow the preference model induces near-optimal performance more often than learning from the control condition's dataset with the same preference model. This observation holds true when learning from the $P_{regret}$-TRAINED-DIFF-DOMAIN dataset, despite no statistically significant difference between the likelihood of that dataset and the TRAINED-DIFF-DOMAIN-Control dataset under the regret preference model. We therefore show that training subjects to follow a preference model in one domain and then eliciting their preferences in a different, downstream domain can improve the alignment of the induced reward function in the downstream domain.

For some domains, it may be difficult or time consuming to teach humans about a specific preference model. Concepts such as the partial return of a trajectory segment may

Figure 10: For the TRAINED-DIFF-DOMAIN experiment, mean cross-entropy loss over each condition's preference dataset with respect to the target preference model. A Wilcoxon paired signed-rank test results in $p < 0.01$ for only the $P_{\Sigma_r}$-TRAINED-DIFF-DOMAIN condition. Note that the loss for the control condition under the regret preference model is already relatively low, leaving less room for improvement.

be difficult for humans to comprehend in other environments, for example when eliciting preferences over large language model outputs. Instead, we change only the wording of the question asked during preference elicitation, investigating how to align human preferences with a specific preference model without relying on their explicit understanding of the preference model.

This experiment consists of three conditions; the $P_{\Sigma_r}$-QUESTION and $P_{regret}$-QUESTION conditions, where we change the preference elicitation instruction in favor of the partial return preference model and regret preference model, respectively, and the QUESTION-Control condition where we use a preference elicitation question that does not seek to influence human preferences towards any specific preference model. Subjects' training matches that of the TRAINED-control condition, and each condition differs only in the wording of the question we ask when eliciting preferences from subjects. We collect data from 9 subjects per condition who are assigned to conditions via random assignment. We replace the data of subjects that are removed due to poor task comprehension or inattentiveness. A video walk-through of the interface used for the $P_{\Sigma_r}$-QUESTION condition is available here, for the $P_{regret}$-QUESTION condition here, and for the QUESTION-Control condition here. Appendix G contains further details.

**Intervention details: Preference elicitation questions** The subjects in each condition see the corresponding question below:

- QUESTION-Control - "Which path do you prefer?", chosen to reduce influence

- $P_{\Sigma_r}$-QUESTION - "Which path has better immediate outcomes?", chosen to focus subjects only on the reward within a segment

- $P_{regret}$-QUESTION - "Which path reflects better decision-making?", chosen to reflect regret's measurement of a segment's deviation from optimality

## 5.3  QUESTION experiment

**Hypothesis 1:** Changing the preference elicitation question in favor of a specific preference model will influence the human to give preferences that are more in conformance with that model.

Results are shown in Figure 12, supporting Hypothesis 1 with a relatively small effect size. The loss for the $P_{\Sigma_r}$-QUESTION dataset under the partial return preference model is lower than the loss for the QUESTION-Control dataset, indicating a small shift in subjects' preferences towards the partial return preference model. A Wilcoxon paired signed-rank test between the likelihoods of these two datasets indicates a statistically significant difference at $p < 0.05$. The $P_{regret}$-QUESTION dataset's loss under the regret preference model is also lower than, but close to, that of the QUESTION-Control's dataset with no statistically significant difference. We observe a similar pattern when looking at the accuracy of the noiseless target preference model over the $P_{\Sigma_r}$-QUESTION and $P_{regret}$-QUESTION datasets, with similar significance. See Appendix I and K for more details.

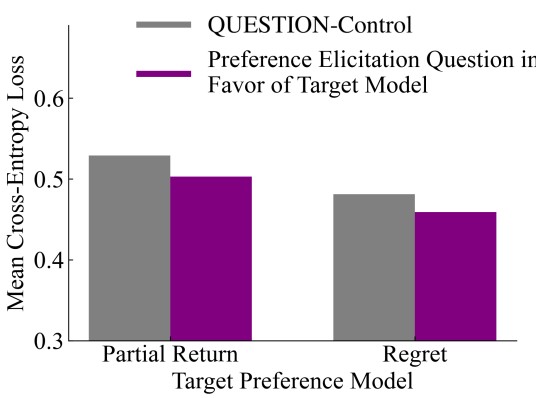

Figure 12: For the QUESTION experiment, mean cross-entropy loss over each condition's preference dataset with respect to the target preference model. Lower is better. Performing a Wilcoxon paired signed-rank test results in significance for only the $P_{\Sigma_r}$-QUESTION condition ($p < 0.05$).

**Hypothesis 2:** Changing the preference elicitation question in favor of a specific preference model leads to learning more aligned reward functions with that preference model.

Figure 13 provides evidence in support of Hypothesis 2. Modifying the preference elicitation question to steer human preferences towards a specific preference model results in a dataset that induces near-optimal behavior more often—when learned from using the target preference model—compared to the control condition dataset. Broadly speaking, this suggests that the question we ask subjects when labeling preferences can effect the performance of the resulting learned reward function.

### 5.3.1  QUESTION-STOCHASTIC-MDP: eliciting preferences in a domain with stochastic transition dynamics

In the QUESTION experiment, modifying the preference-elicitation question shifted subjects' preferences toward the target preference model in a domain with deterministic transitions, albeit with a relatively small effect size. In this experiment, we investigate the same intervention in a domain with stochastic transition dynamics to understand whether people are more susceptible to this intervention in such settings.

Specifically, we use the same grid-world environment as in the QUESTION experiment, but introduce *tunnel* states. When the agent enters a tunnel state, it is teleported to a uniformly random tunnel state elsewhere on the grid. Teleportation incurs no gas cost. As a result, entering a tunnel is optimal when the agent is far from the positive terminal state (the red inverted teardrop), since it offers a chance to rapidly reduce the distance to the goal. However, when the agent is already close to the terminal state, entering a tunnel is suboptimal, as teleportation may move the agent farther away. The board layout used to collect preferences in this stochastic domain is shown in Figure 14. We call this follow-up the QUESTION-STOCHASTIC-MDP experiment.

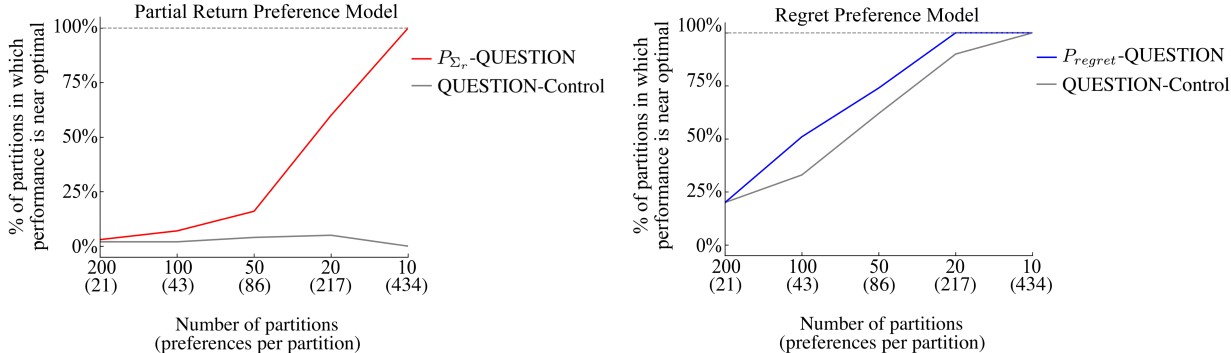

Figure 13: Learning a reward function with the partial return preference model (Left) and regret preference model (Right) from the preferences collected in the QUESTION experiment. See Figure 5 for more details on how this figure was generated.

This experiment includes three conditions. In the $P_{\mathrm{regret}}$–QUESTION-STOCHASTIC-MDP condition, we modify the preference elicitation question to favor the regret preference model. Because the environment's transition dynamics are stochastic—unlike in our other experiments—the regret and change-in-expected-return statistics are no longer equivalent. Accordingly, in the $P_{\Delta\text{-expected-return}}$–QUESTION-STOCHASTIC-MDP condition, the elicitation question is designed to favor the change-in-expected-return preference model. Finally, in the QUESTION-STOCHASTIC-MDP-Control condition, no attempt is made to steer subjects toward any particular preference model. In each condition, subjects are presented with the corresponding question shown below.

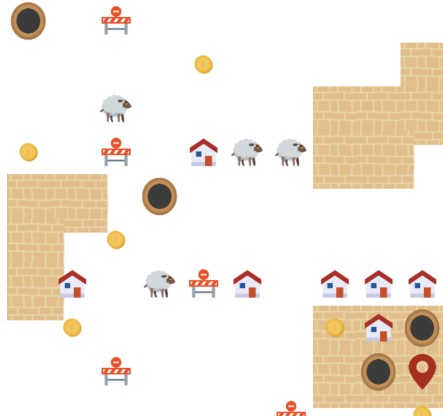

Figure 14: The task shown to human subjects for gathering preferences in the QUESTION-STOCHASTIC-MDP experiment. Brown circles are the tunnels that, when entered, teleport the agent to a uniformly random tunnel state elsewhere on the grid with no additional cost. Entering the two left-most tunnels is optimal, and entering the two right-most tunnels is suboptimal.

- QUESTION-STOCHASTIC-MDP-Control - "Which path do you prefer?", chosen to reduce influence

- $P_{regret}$-QUESTION-STOCHASTIC-MDP - "Which path reflects better decision-making?", chosen to reflect regret's measurement of a segment's deviation from optimality

- $P_{\Delta-\text{expected-return}}$-QUESTION-STOCHASTIC-MDP - "Which path do you prefer to start with?", chosen to focus subjects on both where the vehicle ends up after the segment and the reward within the segment.

We collect data from 10 subjects per condition who are assigned to conditions via random assignment. All experimental details are the same as for the QUESTION condition unless stated otherwise. The subject training interfaces for the QUESTION-STOCHASTIC-MDP experiment exactly match those of the QUESTION experiment, except the QUESTION-STOCHASTIC-MDP experiment uses the modified domain with stochastic transition dynamics.

**Hypothesis 1:** Changing the preference elicitation question in favor of a specific preference model will influence the human to give preferences that are more in conformance with that model.

Hypothesis 1 is supported, though the observed effect size is small (Figure 16) and not statistically significant. The loss of the $P_{\mathrm{regret}}$–QUESTION-STOCHASTIC-MDP dataset is lower under the regret preference model than that of the control condition, and similarly, the loss of the $P_{\Delta\text{-expected-return}}$–QUESTION-STOCHASTIC-MDP dataset is lower under the change-in-expected-return preference model than that of the control condition.

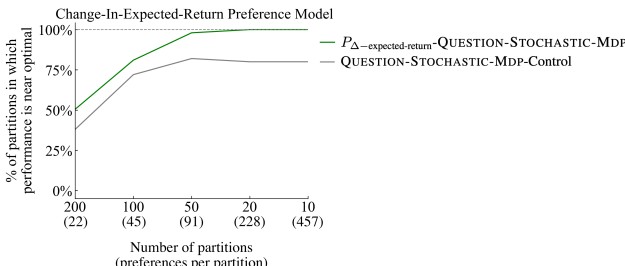 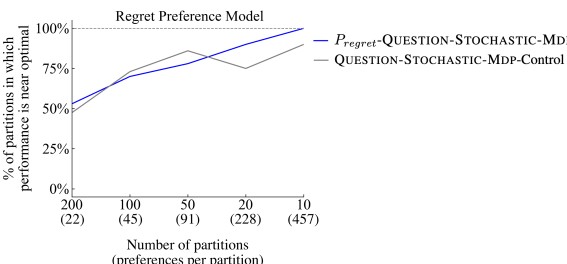

Figure 15: Learning a reward function with the change-in-expected-return preference model (Left) and regret preference model (Right) from the preferences collected in the QUESTION-STOCHASTIC-MDP experiment. See Figure 5 for more details on how this figure was generated.

However, these reductions are not statistically significant according to a paired Wilcoxon signed-rank test at the $p < 0.05$ level.

Interestingly, there is a statistically significant difference between the loss of the $P_{\Delta\text{-expected-return}}$–QUESTION-STOCHASTIC-MDP dataset and the QUESTION-STOCHASTIC-MDP-Control dataset under the regret preference model at $p < 0.05$. This intervention intended to steer participants toward the change-in-expected-return preference model, which it did, but without significance. Yet it also significantly shifted people's preferences toward the regret preference model, an unintended effect. Although Hypothesis 1 is only weakly supported, this result demonstrates that modifying the preference elicitation question can significantly influence participants' preferences towards the regret preference model—albeit in this experiment not towards the preference model that was intended. An exploration of other preference elicitation questions may yield greater effect sizes. Additional details, including the loss of each dataset under the non-target preference model, are provided in Appendix J.

**Hypothesis 2:** Changing the preference elicitation question in favor of a specific preference model leads to learning more aligned reward functions with that preference model in a domain with stochastic transition dynamics.

Hypothesis 2 is only partially supported, as shown in Figure 15. Learning from the $P_{\Delta\text{-expected-return}}$–QUESTION-STOCHASTIC-MDP dataset with the change-in-expected-return preference model induces near-optimal performance more often than learning from the QUESTION-STOCHASTIC-MDP-Control dataset with the same preference model. Learning from the $P_{\text{regret}}$–QUESTION-STOCHASTIC-MDP dataset with the regret preference model outperforms learning from the QUESTION-STOCHASTIC-MDP-Control dataset at larger dataset sizes, and matches or under performs at smaller dataset sizes.

These findings demonstrate that modifying the preference elicitation question in favor of change-in-expected-return improved the alignment of the learned reward function

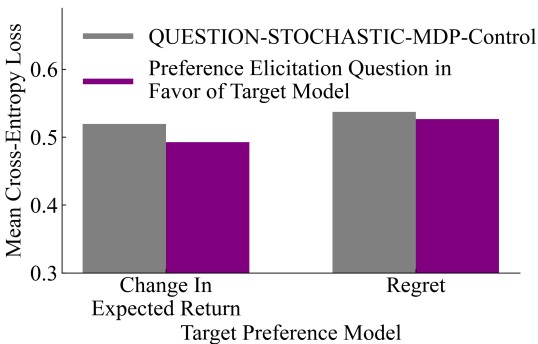

Figure 16: For the QUESTION-STOCHASTIC-MDP experiment, mean cross-entropy loss over each condition's preference dataset with respect to the target preference model. Lower is better. Performing a Wilcoxon paired signed-rank test results in no statistical significance at $p < 0.05$ for either condition.

when using the change-in-expected-return preference model for reward learning. Modifying the preference elicitation question in favor of regret did not produce consistently more aligned reward functions. Although we carefully chose the preference elicitation questions, other phrasings may be more successful at influencing human preferences.

In Table 1 we present a summary of the results for all experiments and interventions considered.

Table 1: Summary of results across all experiments. Each row compares an intervention to its corresponding control under the target preference model. H1 tests whether the intervention shifts human preferences toward the target model, measured by cross-entropy (CE) loss and noiseless accuracy; H2 tests whether this shift improves alignment of the learned reward function. Symbols: $\checkmark^{**}$ = effect in hypothesized direction ($p < 0.01$), $\checkmark^{*}$ = $p < 0.05$, $\checkmark$ = non-significant trend in hypothesized direction, $\approx$ = no meaningful difference, $\checkmark$ = opposite direction. H1 statistical tests: Mann–Whitney $U$ (PRIVILEGED, unpaired) and Wilcoxon signed-rank (others, paired); accuracy uses Fisher's exact test (PRIVILEGED) and Wilcoxon (others). H2 assessments are qualitative, based on near-optimal performance curves.

| Experiment | Target pref. model | $N$ subjects (interv./ctrl) | H1: CE | H1: Acc. | H2: Rew. learn. |
|---|---|---|---|---|---|
| PRIVILEGED | Partial return | 39 / 42 | $\checkmark^{**}$ | $\checkmark^{**}$ | $\checkmark$ |
| | Regret | 43 / 42 | $\checkmark^{**}$ | $\checkmark^{**}$ | $\checkmark$ |
| TRAINED | Partial return | 10 / 10 | $\checkmark^{**}$ | $\checkmark^{**}$ | $\checkmark^{\dagger}$ |
| | Regret | 10 / 10 | $\checkmark^{**}$ | $\checkmark^{**}$ | $\checkmark$ |
| TRAINED-DIFF-DOMAIN | Partial return | 10 / 10 | $\checkmark^{**}$ | $\checkmark^{**}$ | $\checkmark$ |
| | Regret | 10 / 10 | $\approx$ | $\approx$ | $\checkmark$ |
| QUESTION | Partial return | 9 / 9 | $\checkmark^{*}$ | $\checkmark^{*}$ | $\checkmark$ |
| | Regret | 9 / 9 | $\checkmark$ | $\checkmark$ | $\checkmark$ |
| QUESTION-STOCHASTIC-MDP | Change-in-exp.-return | 10 / 10 | $\checkmark^{\ddagger}$ | $\checkmark$ | $\checkmark$ |
| | Regret | 10 / 10 | $\checkmark$ | $\checkmark$ | mixed |

$^{\dagger}$ Poor reward learning attributed to partial return identifiability issues; see Appendix L.6.
$^{\ddagger}$ This intervention also significantly shifted preferences toward the regret model ($p < 0.05$), an unintended effect.

## 6 Conclusion

The choice of preference model used by an RLHF algorithm introduces a source of misalignment between how humans are assumed to generate preferences and how they actually generate preferences, potentially limiting the alignment of the learned reward function. This gap arises because RLHF algorithms must interpret expressed preferences through a particular preference model, even though humans may translate their latent reward judgments into preferences in different ways. Further, even if we could *perfectly* model *all* human preferences, we may wish that preferences are generated by a different model that is computationally efficient for RLHF or provides certain theoretical guarantees. To this end, we propose influencing human preferences towards a chosen preference model through user-interface design—a novel direction for RLHF research. By shaping how humans express preferences during elicitation, our approach reduces the gap between the preference model assumed by the algorithm and the preferences that humans provide.

We focus on influencing people towards three preference models: the change-in-expected-return, regret, and partial return preference models. The latter two are common in prior RLHF work. We first establish that humans can be significantly influenced towards a specific preference model when privileged information about that model is shown during preference elicitation.

We then introduce two practical interventions. In the TRAINED experiment, training subjects to follow a preference model significantly shifts their preferences toward that model in the same domain they were trained in. In a follow-up experiment, training people to follow the partial return preference model in one domain significantly influenced their preferences towards that model in a different, new domain. Training subjects to follow the regret preference model, however, did not influence people towards that model in the new domain, which we attribute to the high cognitive load of learning to explicitly compute the regret. All training-based interventions resulted in learning more aligned reward functions. These experiments highlight

that training subjects to follow a preference model is an effective alignment tool, and that careful subject training interfaces are crucial to avoid fatiguing subjects during preference elicitation.

In the QUESTION experiment, modifying the preference elicitation question shifted participants' preferences toward each of the target preference models in both stochastic and deterministic MDPs. However, these shifts were statistically significant only for the partial return preference model in deterministic environments, and for the regret preference model in stochastic environments under the intervention that was intended to shift preferences towards the change-in-expected-return model. Modifying the preference elicitation question also improved the alignment of the learned reward function when influencing people towards either preference model in deterministic settings, and towards the change-in-expected-return preference model in the stochastic setting. Thus, altering the elicitation question alone—an intervention that is especially easy to implement—can meaningfully shift human preferences and improve the alignment of the learned reward function, although the effects depend on the environment and preference model. This result highlights that even minimal interface choices can influence how humans map their latent reward judgments into expressed preferences.

Results for all interventions are summarized in Table 1. Our findings suggest that human training and preference elicitation interfaces should be viewed as essential tools for improving alignment in RLHF. Notably, the TRAINED and QUESTION experiments offer a viable path forward by influencing human preferences towards a specific preference model to learn more human-aligned reward functions, and they establish a foundation for exploring other effective methods to influence human preferences and extend this methodology to real-world domains. Our results further highlight that the interface and training procedures used to elicit these judgments can systematically influence how preferences are expressed, and therefore which aspects of human reward functions become legible to the learning algorithm.

Several potential future directions follow from this work. First, future research should explore similar interventions in more complex domains such as robotics or embodied agents, where preference-based reinforcement learning is already widely used. Second, there are likely many effective interventions beyond those studied here, including richer training procedures, alternative feedback formats, or interactive preference elicitation interfaces. Finally, it may be possible to deliberately collect multiple preference datasets that correspond to different preference models, allowing learning algorithms to explicitly extract complementary information that a single preference model cannot capture alone. For example, the partial return model informs us that winning a lottery is better than losing, while the regret model informs us that it is suboptimal to buy a lottery ticket; the union of these two models provides more information than either alone. Together, these directions suggest that interface design and human training are promising and underexplored tools for improving the compatibility between human feedback and the assumptions of RLHF algorithms.

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

## A   On the Cognitive Cost of the TRAINED Experiment

The TRAINED intervention is cognitively expensive, as it requires teaching subjects to explicitly compute a segment statistic and label preferences accordingly. This cost is most apparent in the TRAINED-DIFF-DOMAIN experiment, where training toward regret—unlike training toward the cheaper-to-compute partial return—failed to significantly shift preferences in the new domain; subjects in the regret condition also reported significantly more difficulty focusing (mean 2.4 vs. 1.2, $p < 0.05$) and lower confidence computing the statistic (mean 4.6 vs. 6.0, $p < 0.05$, Appendix F.3). The cost may sometimes be justified, since it is a one-time per-annotator cost that, as we show, can produce datasets that yield more aligned reward functions. Moreover, there are other preference models that are less cognitively demanding to follow (e.g., the partial return preference model), and future work can explore how to reduce the cognitive load of learning to follow the regret preference model: since subjects already tend toward regret intuitively in our control conditions, training that targets fast, intuitive (System 1) reasoning rather than explicit computation may achieve comparable conformance at lower cost—an important direction for future work.

## B   The regret preference model

### B.1   Intuition behind the regret preference model

Regret quantifies the extent to which a segment diminishes expected return from $V_{\tilde{r}}^*(s_0^\sigma)$. An optimal segment $\sigma^*$ has 0 regret, while a suboptimal segment $\sigma^{\neg*}$ has positive regret. When two segments have deterministic transitions, end in terminal states, and share the same starting state, this regret preference model is equivalent to the partial return preference model: $P_{regret}(\cdot|\tilde{r}) = P_{\Sigma_r}(\cdot|\tilde{r})$. Conceptually, the partial return preference model assumes that preferences are determined solely by the reward-yielding outcomes *within* the segments, whereas the regret preference model bases preferences on how much the segments deviate from optimal behavior.

### B.2   Comparing two segments with the same start state

When computing the difference in regret for two segments with the same start state, the start state value, $V_{\tilde{r}}^*(s_0^\sigma)$, cancels out:

$regret(\sigma_1|\tilde{r}) - regret(\sigma_2|\tilde{r})$

$= V_{\tilde{r}}^*(s_0^{\sigma_{\sigma_1}}) - (\Sigma_{\sigma_{\sigma_1}}\tilde{r} + V_{\tilde{r}}^*(s_{|\sigma_{\sigma_1}|}^{\sigma_{\sigma_1}})) - V_{\tilde{r}}^*(s_0^{\sigma_{\sigma_2}}) + (\Sigma_{\sigma_{\sigma_2}}\tilde{r} + V_{\tilde{r}}^*(s_{|\sigma_{\sigma_2}|}^{\sigma_{\sigma_2}}))$

$= -(\Sigma_{\sigma_{\sigma_1}}\tilde{r} + V_{\tilde{r}}^*(s_{|\sigma_{\sigma_1}|}^{\sigma_{\sigma_1}})) + (\Sigma_{\sigma_{\sigma_2}}\tilde{r} + V_{\tilde{r}}^*(s_{|\sigma_{\sigma_2}|}^{\sigma_{\sigma_2}}))$

## C   Additional information on the delivery domain and creating a human-labeled preference dataset

When teaching subjects about the delivery domain and constructing the preference datasets for the PRIVILEGED experiment, detailed in Section 5.1, we follow the same procedure as Knox et al. (2022). For the TRAINED and QUESTION experiments, detailed in Sections 5.2 and 5.3 respectively, we modify the interface, preference elicitation, and human subject filtering procedure. These changes are detailed below where applicable. For the follow up experiments—the TRAINED-DIFF-DOMAIN andQUESTION-STOCHASTIC-MDP experiments—all data collection details remain the same as for the TRAINED and QUESTION experiments. The domain used for the TRAINED-DIFF-DOMAIN experiment is outlined separately in Appendix F.

The PRIVILEGED experiment was designed as a proof-of-concept; that human preferences could be influenced towards a specific preference model. When designing the TRAINED and QUESTION experiments, which focus on interventions for the real world, we departed from the experimental protocol used by Knox et al. (2022) to better reflect the hypotheses we wished to test.

### C.1 The delivery domain and task

For the PRIVILEGED, TRAINED, and QUESTION experiments, the delivery domain is structured as a grid composed of cells, each containing a specific type of road surface. The transition dynamics for this domain are deterministic. A task within the delivery domain is illustrated in Figure 3. The state of the delivery agent is its location on the grid. The agent can move one cell in any of the four cardinal directions. Episodes conclude either successfully at the destination, earning a reward of +50, or in failure upon encountering a sheep, resulting in a reward of -50. Non-terminal transitions have a reward equal to the sum of their components: cells with a white road surface carry a -1 reward component, while cells with a brick surface carry a -2 component. Additionally, cells may contain a coin (+1) or a roadblock (-1). Coins remain in place and can, at best, cancel out the cost of the road surface.

Actions that would result in the agent moving into a house or beyond the grid's boundaries result in no movement. In such cases, the reward reflects the current cell's surface component but excludes any coin or roadblock components. The start state distribution, $D_0$, is uniformly random over non-terminal states.

This domain was intentionally designed to make it easy for subjects to recognize poor behavior while making it challenging to discern optimal behavior from most states, mirroring many real-world tasks. This complexity means that some assumptions of the regret preference model, specifically that humans will always prefer optimal segments over suboptimal ones, are not always met, providing a robust test of the model's performance under realistic conditions.

### C.2 Selecting segment pairs for preference elicitation

During the main preference elicitation portion of all experimental conditions, preferences are collected over trajectory segments sampled from the delivery task shown in Figure 3. Below we outline our methodology for selecting segment pairs for labeling in the PRIVILEGED experiment, as well as separately for the TRAINED and QUESTION experiments.

**PRIVILEGED Experiment** We followed the methodology of Knox et al. (2022) for collecting segment pairs, which involved two stages of data collection with differing goals. The first stage sought to characterize human preferences over a range of possible behaviors, including those that would highlight the differences between partial return and regret. The second stage sought to collect preferences over segment pairs that resolve the identifiablity issues of the partial return preference model related to a constant shift in the reward function. We refer readers to Knox et al. (2022) for a detailed description on how these segment pairs were constructed. Figure 17 plots the coordinates from which segment pairs where sampled for each condition in the first stage of data collection. Figures 18 and 19 plot these coordinates for the second of data collection.

The first stage of data collection resulted in $1,359$ segment pairs from 39 subjects for the $P_{\Sigma_r}$-PRIVILEGED condition, $1,418$ segment pairs from 42 subjects for the PRIVILEGED-Control condition, and $1,501$ segment pairs from 43 subjects for the $P_{regret}$-PRIVILEGED condition. All trajectory segments consisted of 3 actions, and the start state for each segment in a pair was different. The second stage of data collection resulted in $1,173$ segment pairs from 25 subjects for the $P_{\Sigma_r}$-PRIVILEGED condition, 375 segment pairs from 8 subjects for the PRIVILEGED-Control condition, and $1,030$ segment pairs from 22 subjects for the $P_{regret}$-PRIVILEGED condition. For each segment pair in the second stage, the agent in one segment takes 3 actions while in the other segment it reaches a terminal state in fewer than 3 actions. Each subject is asked to label preferences for between 35 and 50 segment pairs. No two subjects see the same segment pairs.

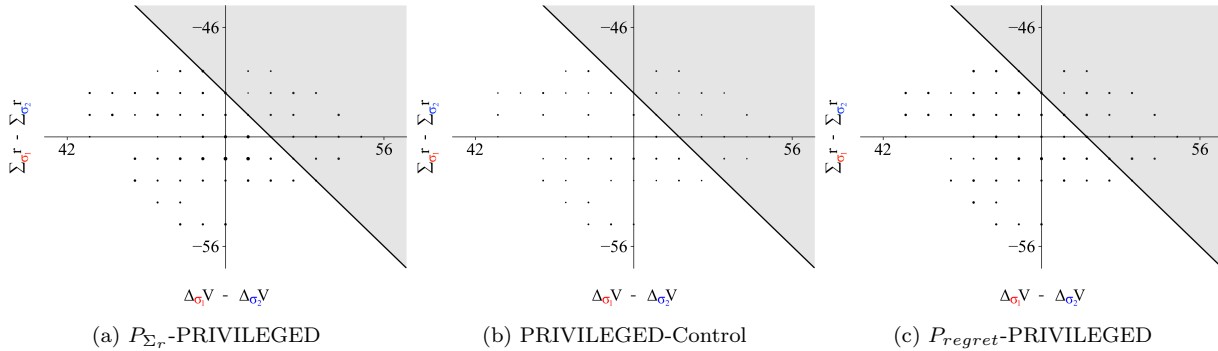

(a) $P_{\Sigma_r}$-PRIVILEGED  (b) PRIVILEGED-Control  (c) $P_{regret}$-PRIVILEGED

Figure 18: The coordinates of the segment pairs shown to subjects for preference labeling in the second stage of data collection for the PRIVILEGED experiment. Each segment pair belonging to these graphs contain one segment where the agent terminates at a positive terminal state and one where it does not. The proportionality of the circles are consistent across this plot and the 3 subplots of Figure 17 and 19.

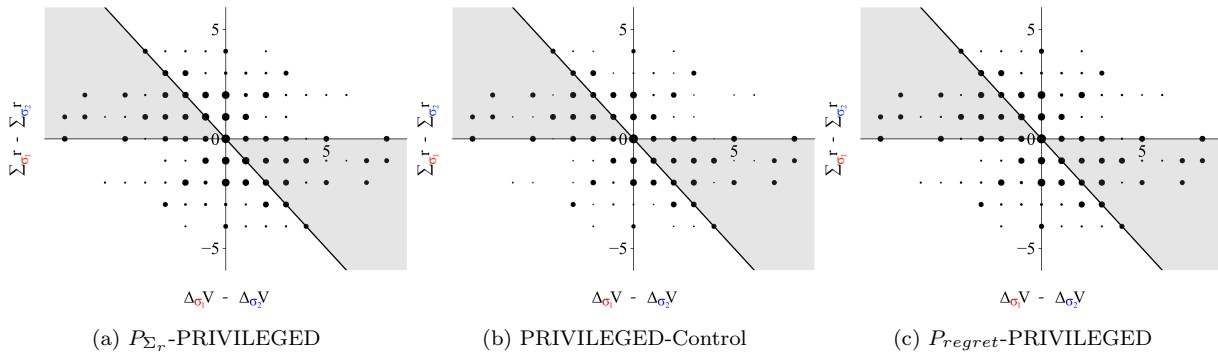

(a) $P_{\Sigma_r}$-PRIVILEGED  (b) PRIVILEGED-Control  (c) $P_{regret}$-PRIVILEGED

Figure 17: The coordinates of the segment pairs shown to subjects for preference labeling in the first stage of data collection for the PRIVILEGED experiment. The $x$-axis is the difference in the change in state value between the two segments and the $y$-axis is partial return differences between the two segments. The areas of the circles are proportional to the number of segment pairs at that point. The proportionality is consistent across this plot and the 3 subplots of Figures 18 and 19.

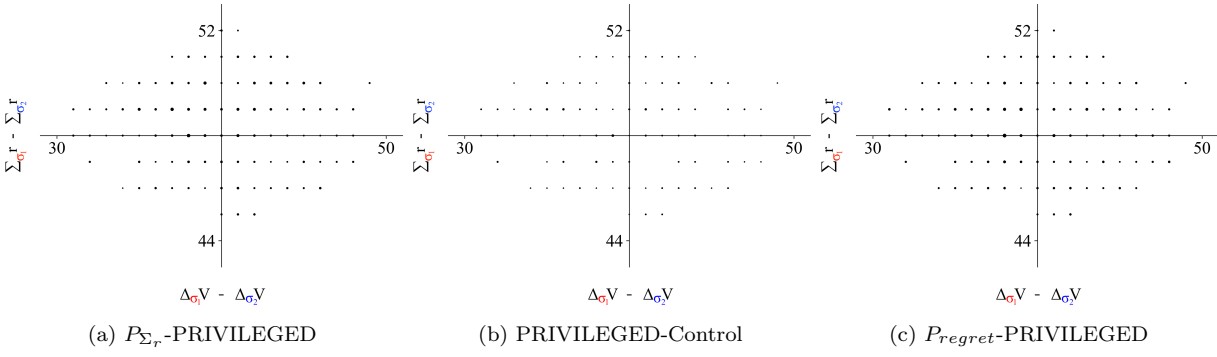

(a) $P_{\Sigma_r}$-PRIVILEGED  (b) PRIVILEGED-Control  (c) $P_{regret}$-PRIVILEGED

Figure 19: The coordinates of the segment pairs shown to subjects for preference labeling in the second stage of data collection for the PRIVILEGED experiment. Each segment pair belonging to these graphs contain one segment where the agent terminates and one where it does not. The proportionality of the circles are consistent across this plot and the 3 subplots of Figure 17 and 18.

**TRAINED and QUESTION Experiments**   Subjects label the same dataset of 500 segment pairs for each of the 3 conditions in the TRAINED and QUESTION experiment. We chose 500 segment pairs for preference labeling by splitting the $P_{\Sigma_r}$-PRIVILEGED, PRIVILEGED-Control, and $P_{regret}$-PRIVILEGED

datasets into different numbers of same sized partitions. We then identified the smallest partition size where the likelihood of the influenced preference dataset was always significantly higher than that of the control condition, defined as being 100 times more likely.

In 72 of the 500 pairs used for labeling, the agent in one trajectory segment takes 3 actions, while in the other segment, it reaches a terminal state in fewer than 3 actions. Knox et al. (2022) found that these trajectory segments are essential for learning a reward function with the partial return preference model in this domain. The remaining trajectory segments consist of 3 actions sampled uniformly randomly from all possible actions. The trajectory segments in a segment pair have the same start state. The order of trajectory segments within a pair and the order of segment pairs shown to subjects is uniformly random. Each subject is asked to label preferences for 50 segment pairs. No two subjects within a condition see the same segment pairs, except for the last segment pair used to test for task comprehension and attention. Figure 25 illustrates the segment pairs sampled for labeling for all experimental conditions within these two experiments.

In the TRAINED experiment we collected data from 10 subjects per condition. In the QUESTION experiment, we sought to do the same but ran into the following issue: We randomly sample subjects to complete our study from a standard sample of available subjects who meet certain criteria (see Appendix C.3), utilize random assignment to assign each subject to a condition, and recollect data removed from subjects who failed the comprehension tests (detailed in Appendix C.4). For the $P_{regret}$-QUESTION condition, we were unable to find a 10th subject who passed the comprehension test before the subject sample population potentially changed significantly over time. As such, we were no longer confident that we could claim the subjects from all conditions in the QUESTION experiment were drawn from the same population. Therefore, we collected data from only 9 subjects per condition in the QUESTION experiment, resulting in a maximum dataset size of 450 preferences per condition.

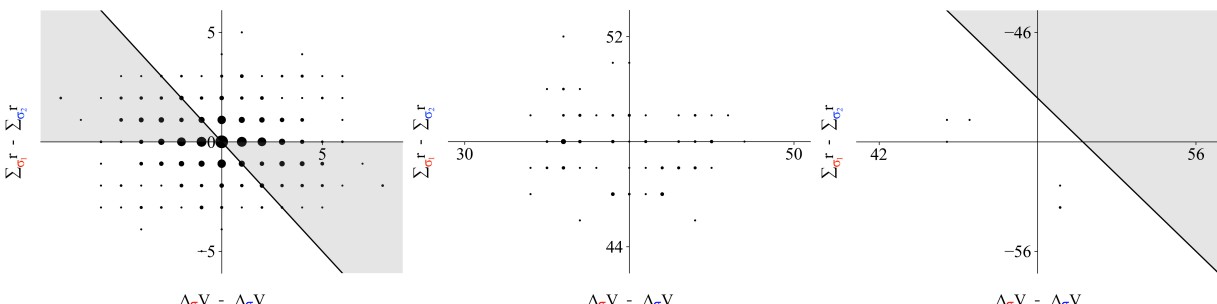

Figure 20: The coordinates of the segment pairs shown to subjects for preference labeling for all conditions for the TRAINED and QUESTION experiments. The areas of the circles are proportional to the number of segment pairs at that point. The proportionality is consistent across this plot.

### C.3 Recruiting human subjects

All subject compensation amounts were chosen using the median time subjects took during a pilot study and then calculating the payment to result in $15 USD per hour. This hourly rate of $15 was chosen because it is commonly recommended as an improved US federal minimum wage.

**PRIVILEGED Experiment**   We recruited subjects with IRB approval via Amazon Mechanical Turk and paid subjects $5 per experiment. Subjects had to be located in the United States, have an approval rating of at least 99%, and have completed at least 100 other studies on Mechanical Turk to join our study. Due to an experimental error, we did not show the IRB-approved consent form to participants after they accepted our study on Mechanical Turk. We reported this issue to our IRB and received approval to use the collected data.

**TRAINED and QUESTION Experiments**   We recruited subjects with IRB approval via Prolific. We paid subjects in the TRAINED experiment (see Section 5.2) $7.50 for completing the study. We observed that subjects took about half the time to complete the study when they were not taught about a specific preference model, such as in the control condition, and so for the QUESTION experiment (see Section 5.3)

we paid subjects \$3.75 for completing the study. Subjects were recruited via Prolific from a standard sample, and were required to be both fluent in English and located in the United States.

### C.4 Filtering subject's data

We evaluated each subject's understanding of the delivery domain and excluded those who lacked sufficient understanding. Participants were required to complete a task-comprehension survey, from which we derived a task-comprehension score. The questions and corresponding answer choices are detailed in Table 2. Participants received 1 point for fully correct answers and 0.5 points for partially correct answers. For the PRIVILEGED experiment—and separately for the TRAINED and QUESTION experiments—the threshold score for removing worker data was determined by visually inspecting a histogram of the scores, aiming to strike a balance between upholding high comprehension standards and retaining a sufficient dataset for analysis.

In addition to filtering based on the task-comprehension survey, we also removed data from any participant who ever preferred a segment where the agent ends in a negative terminal state—the worst possible outcome—over a segment where the agent does not.

**PRIVILEGED Experiment** Subjects could achieve a score on the task-comprehension survey ranging from 0 to 7. Data from participants scoring below 4.5 was discarded. All subjects were shown at least one segment pair containing a segment where the agent ends in a negative terminal state and a segment where it does not, and their data is removed if they prefer the former. These segment pairs are used to test for task comprehension and attentiveness. Because of a data management error, the filtered-out data was lost and we don't otherwise know how many subjects were filtered out for this expirement.

**TRAINED and QUESTION Experiment** Subjects could achieve a score on the task-comprehension survey ranging from 0 to 6. Data from participants scoring below 3.5 was discarded. The last segment pair shown to participants during preference elicitation always contained a segment where the agent ends in a negative terminal state and a segment where it does not. Other segment pairs shown to subjects may also illustrate this scenario.

Figure 21: An episode from the grid-world domain used for teaching subjects about the domain transition and reward function, as well as to aid in their understanding of partial return (i.e., "score") and regret (i.e., "opportunity cost") in the PRIVILEGED condition.

Across all conditions in the TRAINED experiment, the data from 19/49 subjects was removed: 9/19 from the TRAINED-Control condition, 9/19 from the $P_{\Sigma_r}$-TRAINED condition, and 1/11 from the $P_{regret}$-TRAINED condition. The data from 33/60 subjects in the QUESTION experiment was removed: 11/20 from the QUESTION-Control condition, 9/18 from the $P_{\Sigma_r}$-QUESTION condition, and 13/22 from the $P_{regret}$-QUESTION condition.

## D PRIVILEGED Experiment Interface Details

For all conditions in the PRIVILEGED experiment, when subjects interact with episodes from the grid-world domain we display four segment statistics: the "score" or partial return, the "best possible score from start" or $V_r^*(s_0^\sigma)$, the "best possible score given your moves" or $\tilde{r} + V_r^*(s_{|\sigma|}^\sigma)$, and the "opportunity cost" or regret, which is the difference between the previous two components. We explain $V_r^*(s_0^\sigma)$ as "the most money the vehicle could have made from the start", $\tilde{r} + V_r^*(s_{|\sigma|}^\sigma)$ as "the most money the vehicle can make from the start, including the route you've taken so far", and regret as "the difference between the two" and the "minimum

Table 2: The task comprehension survey, designed to test participant's comprehension of the domain for the purpose of filtering data. Each full credit answer earned 1 point; each partial credit answer earned 0.5 points. The left most column indicates the experiment where the given QUESTION was used to filter subjects. We discarded the data of participants who scored less than 4.5 points overall for the PRIVILEGED experiment, and less than 3.5 points overall for the TRAINED and QUESTION experiments.

| Experiment | QUESTION | Full credit answer | Partial credit answer | Other answer choices |
|---|---|---|---|---|
| Privileged, Trained, QUESTION | What is the goal of this world? (Check all that apply.) | • To maximize profit | • To get to a specific location.
• To maximize profit
Partial credit was given if both answers were selected. | • To drive as far as possible to explore the world.
• To collect as many coins as possible.
• To collect as many sheep as possible.
• To drive sheep to a specific location. |
| Trained, QUESTION | What happens when you run into a house? | • You incur a gas cost and don't go anywhere. | • You incur a gas cost and a cost for hitting the house, and you don't go anywhere.
• You incur a gas cost and a cost for hitting the house, and you drive over the house.
• Nothing happens. | • The episode ends.
• You get stuck.
• To collect as many sheep as possible. |
| PRIVILEGED | What happens when you run into a house? (Check all that apply.) | • You pay a gas penalty.
• You can't run into a house; the world doesn't let you move into it.
Full credit was given if both answers were selected. | • You pay a gas penalty.
• You can't run into a house; the world doesn't let you move into it.
Partial credit was given if only one answer was selected. | • The episode ends.
• You get stuck.
• To collect as many sheep as possible. |
| Privileged, Trained, QUESTION | What happens when you run into a sheep? (Check all that apply.) | • The episode ends.
• You are penalized for running into a sheep.
Full credit was given if both answers were selected. | • The episode ends.
• You are penalized for running into a sheep.
Partial credit was given if only one answer was selected. | • You are rewarded for collecting a sheep. |
| Privileged, Trained, QUESTION | What happens when you run into a roadblock? (Check all that apply.) | • You pay a penalty. | | • The episode ends.
• You get stuck.
• You can't run into a roadblock; the world doesn't let you move into it. |
| Privileged, Trained, QUESTION | Is running into a roadblock ever a good choice in any town? | • Yes, in certain circumstances. | | • No. |
| PRIVILEGED | What happens when you go into the brick area? (Check all that apply.) | • You pay extra for gas. | | • The episode ends.
• You get stuck in the brick area.
• You can't go into the brick area; the world doesn't let you move into it. |
| Privileged, Trained, QUESTION | Is entering the brick area ever a good choice? | • Yes, in certain circumstances | | • No |

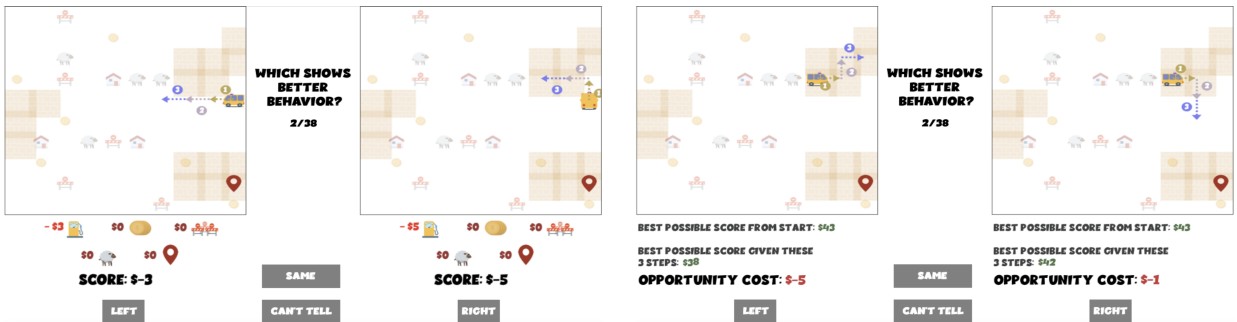

Figure 22: An example of the preference elicitation interface shown to subjects in the $P_{\Sigma_r}$-PRIVILEGED condition (Left) and the $P_{regret}$-PRIVILEGED condition (Right).

amount of money lost by taking your route instead of the best route". See Figure 21 for an example of what humans see when interacting with an episode from the domain.

For all conditions in the PRIVILEGED experiment, we ask subjects to label preferences for $35-50$ segment pairs using the QUESTION "Which shows better behavior?". Only the information shown during preference elicitation differs between conditions, as detailed below.

### D.1  $P_{\Sigma_r}$-PRIVILEGED Condition Interface details

During preference elicitation, we display the "score", or the partial return, for the vehicle's path and each corresponding reward component. See Figure 22 for the preference elicitation interface for the $P_{\Sigma_r}$-PRIVILEGED condition.

### D.2  $P_{regret}$-PRIVILEGED Condition Interface details

During preference elicitation, we display the "best possible score from start" or $V_r^*(s_0^\sigma)$, the "best possible score given your moves" or $\tilde{r} + V_r^*(s_{|\sigma|}^\sigma)$, and the "opportunity cost" or regret, which is the difference between the previous two components. See Figure 22 for the preference elicitation interface for the $P_{regret}$-PRIVILEGED condition.

### D.3  PRIVILEGED-Control Condition Interface details

During preference elicitation, we do not display any segment statistics. See Figure 23 for the preference elicitation interface for the PRIVILEGED-Control condition.

## E  TRAINED Experiment Interface details

The interfaces employed for each condition in the TRAINED experiment differ in what preference model—if any—human subjects' preferences are influenced towards. Therefore, the concepts taught throughout the study and the preference elicitation instruction differ between conditions, as outlined below.

### E.1  $P_{\Sigma_r}$-TRAINED condition Interface details

When subjects interact with episodes from the grid-world domain, we display the "score so far", or the partial return, for the vehicle's path. We explain how the "score so far" is computed, have them compute it for three trajectory segments while providing

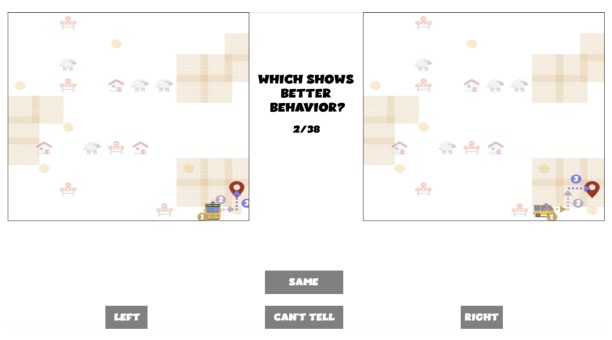

Figure 23: An example of the preference elicitation interface shown to subjects in the PRIVILEGED-Control condition.

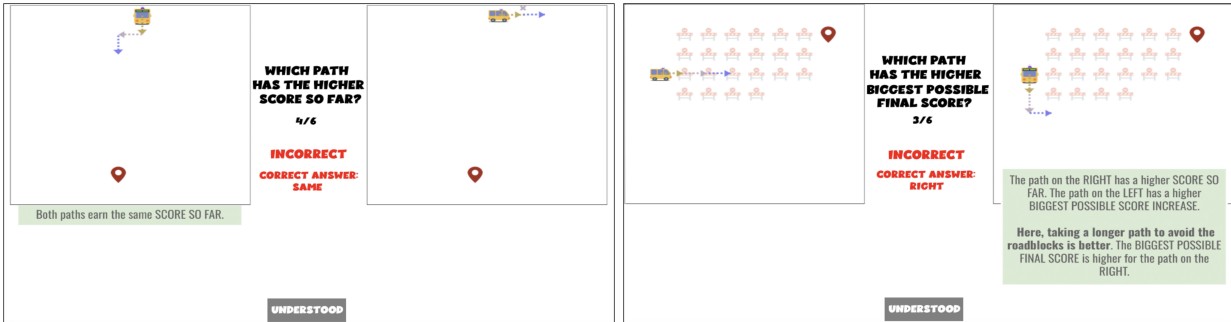

Figure 24: In the TRAINED experiment, subjects are shown two sets of six segment pairs where, after labeling their preference, they are given feedback on its correctness. For these practice segment pairs, subjects are given an explanation of why one segment is preferable to another regardless of whether their preference is correct. The left interface above displays an example of preference feedback for the partial return preference model and the right interface for the regret preference model. Preference feedback is only given for domain tasks separate from the delivery task used during the main preference elicitation session. To avoid technical jargon, we refer to partial return as "score so far", the end state value as "biggest possible score increase", and regret as "biggest possible final score". Subjects are taught these concepts during training.

feedback, and then let them interact with additional episodes to observe the "score so far".

After understanding how to compute partial return, subjects were instructed to use it when generating preferences. Initially, they labeled preferences for six segment pairs without specific guidance. Subsequently, they were told to generate preference labels based on the "score so far" and walked through a detailed example. They then labeled six more segment pairs, receiving feedback on the correctness of their preferences. An example of this interface is shown in Figure 24. Finally, subjects were instructed on how *not* to generate preferences (e.g., "do not select the path where the van looks like it might achieve a higher score in the future"), and given another six segments pairs to label with feedback on their preferences.

The trajectory segments used to teach subjects the partial return preference model were collected from domain maps, excluding the one depicted in Figure 3, which is reserved for the main preference elicitation portion. These pedagogical segment pairs were selected to illustrate scenarios where the partial return of both segments were equal, where one segment had a higher partial return but higher regret than the other (i.e., the two competing preference models would disagree on the preference label), and where one segment had a higher partial return and lower regret than the other. The preferences collected for these segment pairs were not used for reward learning.

After learning to use the partial return preference model, subjects interacted with the map shown in Figure 3 and generated preferences for 50 segment pairs from this task. When labeling these segment pairs, subjects were asked "which path has the highest score so far?". This is the main preference elicitation phase, where no feedback or information about the ground-truth reward function is provided. The preference elicitation interface is shown in Figure 4.

### E.2  $P_{regret}$-**TRAINED condition Interface details**

We progressively teach subjects how to compute the regret of a trajectory segment by sequentially introducing its components. Because the transition dynamics of the MDPs used in this experiment for subject training and preference elicitation are deterministic, we use the following, conceptually simpler formulation of regret:

$$regret(\sigma_t|\tilde{r}) \triangleq V_{\tilde{r}}^*(s_t^\sigma) - [\tilde{r}_t + V_{\tilde{r}}^*(s_{t+1}^\sigma)].$$

Note that in this setting, this formulation of regret is equivalent to that of Equation 3 and to the change in expected return in Equation 4. When subjects interact with episodes from the delivery domain, we display the components of regret as we introduce them. First, we introduce the "score so far," $\Sigma_\sigma r$. Next, we explain the "biggest possible score increase," $V_r^*(s_{|\sigma|}^\sigma)$. Finally, we present the "biggest possible final score," which combines both components as $\Sigma_\sigma r + V_r^*(s_{|\sigma|}^\sigma)$. For each component, we explain how it is computed and allow

subjects to interact with various tasks to observe the corresponding values. Additionally, we ask subjects to compute $V_r^*(s_{|\sigma|}^\sigma)$ and $\Sigma_\sigma r + V_r^*(s_{|\sigma|}^\sigma)$ for three different trajectory segments each, providing feedback on their answers. Since we always present segment pairs that share the same start state, we do not introduce $V_r^*(s_0^\sigma)$ to subjects because this component cancels out when computing preference distributions using the regret preference model (see Appendix B.2).

After understanding how to compute regret, subjects are taught to use it when generating preferences following a procedure similar to that in Section E.1. They are first asked to label preferences for six trajectory segments. Then, they are instructed to generate preference labels based on the "biggest possible final score" and shown a detailed example. Subjects label six more trajectory segments, receiving feedback on the correctness of their preferences as illustrated in Figure 24. Following this, they are instructed on how *not* to generate preferences (e.g., "do not select the path that merely has the higher score so far"). Finally, they are given another six segment pairs to label, with feedback provided on their preferences. Note that these pedagogical segment pairs are the same as those used in the $P_{\Sigma_r}$-TRAINED condition, detailed in Appendix E.1.

The subsequent preference elicitation procedure used to build the dataset of preferences for reward learning is identical to the procedure outlined in Section E.1, except subjects are asked "which path has the highest biggest possible final score" when generating preferences.

### E.3   TRAINED-Control condition Interface details

When subjects interact with episodes from the grid-world domain, we display the "score so far", or the partial return, for the vehicle's path and explain this statistic in the same way as for the $P_{\Sigma_r}$-TRAINED condition. We do not train subjects to compute any segment statistic, nor do we instruct them on how to generate preferences like in the other conditions. During preference elicitation, we ask subjects "Which path do you prefer?", a QUESTION that does not seek to influence subjects towards any preference model. This preference elicitation procedure aims to be generally representative of standard approaches for collecting feedback for RLHF.

### E.4   Testing subjects comprehension of the taught preference model

When teaching subjects to follow a specific preference model, we present them with two sets of six practice segment pairs and provide feedback on their preference labels for these pairs. We test the correlation between the subject's adherence to the taught preference model during the last six practice pairs and in the main preference elicitation portion of the study. We compute the Spearman correlation coefficient between the fraction of human preferences that agree with the noiseless version of the taught model in the last six practice segment pairs and in the fifty segment pairs shown during the main preference elicitation portion.

We are not able to perform this analysis for the $P_{\Sigma_r}$-TRAINED condition; the fraction of human preferences that agree with the noiseless version of the partial return preference model in the last six practice segment pairs remains constant for all subjects and therefore the Spearman correlation coefficient is undefined. For the $P_{regret}$-TRAINED condition, we compute a Spearman correlation coefficient of $-0.137$ with $p = 0.706$. We suspect that the high $p$-value is a result of the small sample size of only 10 subjects.

### E.5   Surveying subject agreement of the taught preference model

During the post-study task-comprehension survey, we assess subjects' personal agreement with the taught preference model. For the $P_{\Sigma_r}$-TRAINED condition, we ask, "We told you that the better path is always the one with the higher SCORE SO FAR. How often did you agree with this?" For the $P_{regret}$-TRAINED condition, we ask, "We told you that the better path is always the one with the higher BIGGEST POSSIBLE FINAL SCORE. How often did you agree with this?" Responses are given on a 7-point Likert scale, where 1 indicates "always disagreed" and 7 indicates "always agreed." The mean response for the $P_{\Sigma_r}$-TRAINED condition was 4.2 with a variance of 1.56, while the $P_{regret}$-TRAINED condition had a mean response of 6.3 with a variance of 1.61. These results suggest that subjects personally aligned more with the regret-based labeling of segment pairs than with the partial return-based approach.

For both the $P_{\Sigma_r}$-TRAINED and $P_{regret}$-TRAINED conditions, we also asked subjects "How helpful were our explanations on why one path was better than another path for your own decision making?" The mean response for the $P_{\Sigma_r}$-TRAINED condition was 5.8 with a variance of 2.36, while the $P_{regret}$-TRAINED condition had a mean response of 6.4 with a variance of 0.84. We interpret these results as general satisfaction with our protocol for teaching subjects about a specific preference model.

### E.6 Number of preferences per collected dataset

We collected 500 preferences for each condition outlined in the TRAINED experiment. Because we discard samples where a subject chose "Can't Tell" instead of a preference, each dataset contains a different number of preferences indicated in Table 3.

## F TRAINED-DIFF-DOMAIN Experiment Interface Details

The data collection methodology of the TRAINED-DIFF-DOMAIN experiment closely follows that of the TRAINED experiment. Unless specified, all details regarding how the preference dataset was constructed match that of the TRAINED experiment in Appendix C.

The procedure executed to train subjects to follow a chosen preference model exactly matches the procedure outlined for the TRAINED experiment in

Table 3: The number of preferences in each preference dataset resulting from the TRAINED experiment.

| Condition | Number of Preferences in Dataset |
|---|---|
| $P_{\Sigma_r}$-TRAINED | 497 |
| TRAINED-Control | 473 |
| $P_{regret}$-TRAINED | 491 |

Appendix E. After the subject-training procedure, we introduce the subjects to a space domain outlined in Section F.1. We teach subjects about the ground-truth reward function of this domain—which differs from the ground-truth reward function of the delivery domain used for teaching the chosen preference model. Subjects control the agent on space domain maps designed to teach one or two concepts at a time. After teaching subjects to understand the space domain and task, we elicit their preferences.

### F.1 Space domain and task

The space domain is structured as a grid composed of cells, each containing a specific type of surface. A task within the delivery domain is illustrated in Figure 9. The state of the delivery agent is its location on the grid. The agent can move one cell in any of the four cardinal directions. Episodes conclude either successfully at the destination, earning a reward of +50, or in failure upon encountering a sheep, resulting in a reward of -100. Non-terminal transitions have a reward equal to the sum of their components: cells with a white road surface carry a -5 reward component, while cells with a asteroid field surface carry a -15 component. Additionally, cells may contain a gold nugget (+3) or space junk (-1). Gold nuggets remain in place.

Actions that would result in the agent moving beyond the grid's boundaries result in no movement. In such cases, the reward reflects the current cell's surface component but excludes any gold nugget or space junk components. The start state distribution, $D_0$, is uniformly random over non-terminal states.

### F.2 Differences between conditions

The $P_{\Sigma_r}$-TRAINED-DIFF-DOMAIN and $P_{regret}$-TRAINED-DIFF-DOMAIN conditions differ as detailed in Appendix E. After human subjects are taught the chosen preference model, they are shown the same content in both conditions to learn about the new space domain.

For the TRAINED-DIFF-DOMAIN-Control condition, we follow the same procedure as the TRAINED-Control condition, except that we teach human subjects about the new space domain after teaching them about the delivery domain. Therefore, for the TRAINED-DIFF-DOMAIN-Control condition, subjects are taught the same two domains as in the other two conditions for this experiment. For all conditions, subject preferences are immediately elicited after they are taught about the space domain.

### F.3 Surveying subjects and evidence of cognitive load

In Section 5.2, we hypothesize that subjects' preferences were not influenced towards the regret preference model in this experiment because of the cognitive load of learning to follow the regret preference model, and then learning about a new domain, fatigued subjects. On the other hand, subjects were influenced towards the partial return preference model in the new domain in this experiment, and towards both preference models when eliciting preferences in the same domain as used for teaching (i.e., in the TRAINED-experiment). Here we provide evidence supporting the hypothesis subjects faced more cognitive strain in the $P_{regret}$-TRAINED-DIFF-DOMAIN condition than the other conditions, which may have impacted their ability to follow the regret preference model.

After preference elicitation, we ask subjects:

- Q1: How difficult was it to focus through the entire study?

- Q2: How good do you think you are at determining the [taught segment statistic]?

- Q3: We told you that the better path is always the one with the higher [taught segment statistic]. How often did you agree with this?

Answers were given on a 7-point Likert scale. For each question, we performed a Mann-Whitney U test on the reported values by the 10 subjects in each condition. For Q1, an answer of 1 indicates "very easy" and an answer of 7 indicates "very difficult". The mean reported value for Q1 was 2.4 for the $P_{regret}$-TRAINED-DIFF-DOMAIN condition and 1.2 for the $P_{\Sigma_r}$-TRAINED-DIFF-DOMAIN condition with $p < 0.05$, indicating a statistically significant difference in subject's self reported ability to focus throughout the study; subjects found it easier to focus in the $P_{\Sigma_r}$-TRAINED-DIFF-DOMAIN condition.

For Q2, an answer of 1 indicates "very bad" and an answer of 7 indicates "very good". The mean reported value for Q2 was 4.6 for the $P_{regret}$-TRAINED-DIFF-DOMAIN condition and 6.0 for the $P_{\Sigma_r}$-TRAINED-DIFF-DOMAIN condition with $p < 0.05$, indicating a statistically significant difference in subject's self reported ability to compute the taught segment statistic; subjects found it easier to compute the taught segment statistic in the $P_{\Sigma_r}$-TRAINED-DIFF-DOMAIN condition.

For Q3, an answer of 1 indicates "always disagreed" and an answer of 7 indicates "always agreed". The mean reported value for Q2 was 6.6 for the $P_{regret}$-TRAINED-DIFF-DOMAIN condition and 4.8 for the $P_{\Sigma_r}$-TRAINED-DIFF-DOMAIN condition with $p < 0.05$, indicating a statistically significant difference in subject's self reported agreement with the taught segment statistic; subjects agreed more with the taught segment statistic in the $P_{regret}$-TRAINED-DIFF-DOMAIN condition.

No statistically significant differences were observed between Q1 and Q2 in the TRAINED experiment where subjects were significantly influenced towards both taught preference models. These results in combination indicate that for the TRAINED-DIFF-DOMAIN condition, subjects found it significantly more difficult to focus and follow the regret preference model than the partial return preference model, despite higher self reported agreement with the regret preference model. Therefore, we hypothesize that teaching subjects about two domains, followed by teaching them about the regret preference model, posed a significant cognitive load that curbed their ability to follow the regret preference model at preference elicitation time. On the other hand, the partial return preference model is comparatively easier to compute and so subjects may have not felt the same fatigue in the $P_{\Sigma_r}$-TRAINED-DIFF-DOMAIN condition. These results generally serve as evidence that teaching subjects to follow a chosen preference model can be an effective way to influence their preferences towards that preference model in a different domain, but careful interface design is important to avoid fatiguing subjects. Otherwise, such interventions may become ineffective.

## G   QUESTION Experiment Interface Details

The conditions in the QUESTION experiment only differ in what question is asked during preference elicitation. The control condition used for the QUESTION experiment is identical to that used for the TRAINED experiment, detailed in Appendix E.3.

### G.1 Choosing the preference elicitation question

The authors of this paper were asked to propose possible questions to ask subjects during preference elicitation that might influence their preferences toward either preference model. The first author selected the three most appealing options for each preference model, each author ranked these options in order of desirability, and then ranked-choice voting was employed to select the winner. For guiding human preferences towards the regret preference model the three options ranked by each author were "Which path shows better decision-making?", "Which path reflects better decision-making?, and "Which path is more likely to be taken by an expert?". For guiding human preferences towards the partial return preference model, the three options were "Which path would be better if the task ended after the path?", "Which path has better immediate outcomes?", and "Which path looks better, considering only exactly what happened during the path?". After ranking these questions by their likely ability to guide human preferences towards the regret or partial return preference models, the second question won 80% and 60% of the time respectively.

### G.2 Number of preferences per collected dataset

Table 4 displays the number of preferences collected for each condition in the QUESTION experiment after discarding samples where a subject chose "Can't Tell" instead of a preference.

## H Correlations between preferences and segment statistics

Recall that we compute the regret of a trajectory segment with deterministic transitions as follows: $regret(\sigma|\tilde{r}) = V_{\tilde{r}}^*(s_0^\sigma) - (\Sigma_\sigma\tilde{r} + V_{\tilde{r}}^*(s_{|\sigma|}^\sigma))$, where one of the 3 components of regret is partial return, $\Sigma_\sigma\tilde{r}$. We combine two components of $regret(\sigma|r)$ to simplify analysis, introducing the following shorthand: $\Delta_\sigma V_{\tilde{r}} \triangleq V_{\tilde{r}}^*(s_{|\sigma|}^\sigma) - V_{\tilde{r}}^*(s_0^\sigma)$.

The change in state value, $\Delta_\sigma V_{\tilde{r}}$, should have a greater effect on human preferences that are more

Table 4: The number of preferences in each preference dataset resulting from the QUESTION experiment.

| Condition | Number of Preferences in Dataset |
|---|---|
| $P_{\Sigma_r}$-QUESTION | 437 |
| QUESTION-Control | 434 |
| $P_{regret}$-QUESTION | 442 |

aligned with the regret preference model, and partial return should have a greater effect on human preferences more aligned with the partial return preference model. The datasets of preferences are visualized in Figure 25. Note that on the diagonal line in Figure 25, $regret(\sigma_2|r) = regret(\sigma_1|r)$, making the $P_{regret}$ preference model indifferent.

Figure 25 shows that, when influencing human preferences towards the regret preference model in the PRIVILEGED and TRAINED experiments, $\Delta_\sigma V_{\tilde{r}}$ has influence on the resulting preference dataset independent of partial return. This is evident for the $P_{regret}$-PRIVILEGED and $P_{regret}$-TRAINED condition's dataset plots when focusing only on points at a chosen $y$-axis value; if the colors along the corresponding horizontal line reddens as the $x$-axis value increases, then $\Delta_\sigma V_{\tilde{r}}$ appears to have independent influence. Visually, $\Delta_\sigma V_{\tilde{r}}$ also exhibits independent influence on the preference datasets from the control condition for all experiments. When influencing human preferences towards the partial return preference model in the PRIVILEGED and TRAINED experiments, $\Delta_\sigma V_{\tilde{r}}$, has significantly less influence on the resulting preference dataset as evident by the $x$-axis–rather than the diagonal line–partitioning most red and blue points in the $P_{\Sigma_r}$-PRIVILEGED and $P_{\Sigma_r}$-TRAINED condition's dataset plots.

Visual inspection leads us to conclude that $\Delta_\sigma V_{\tilde{r}}$ independently influences human preferences for all conditions in the QUESTION experiment, indicating that regardless of the preference elicitation question, the change in state value is still correlated with how subjects label preferences.

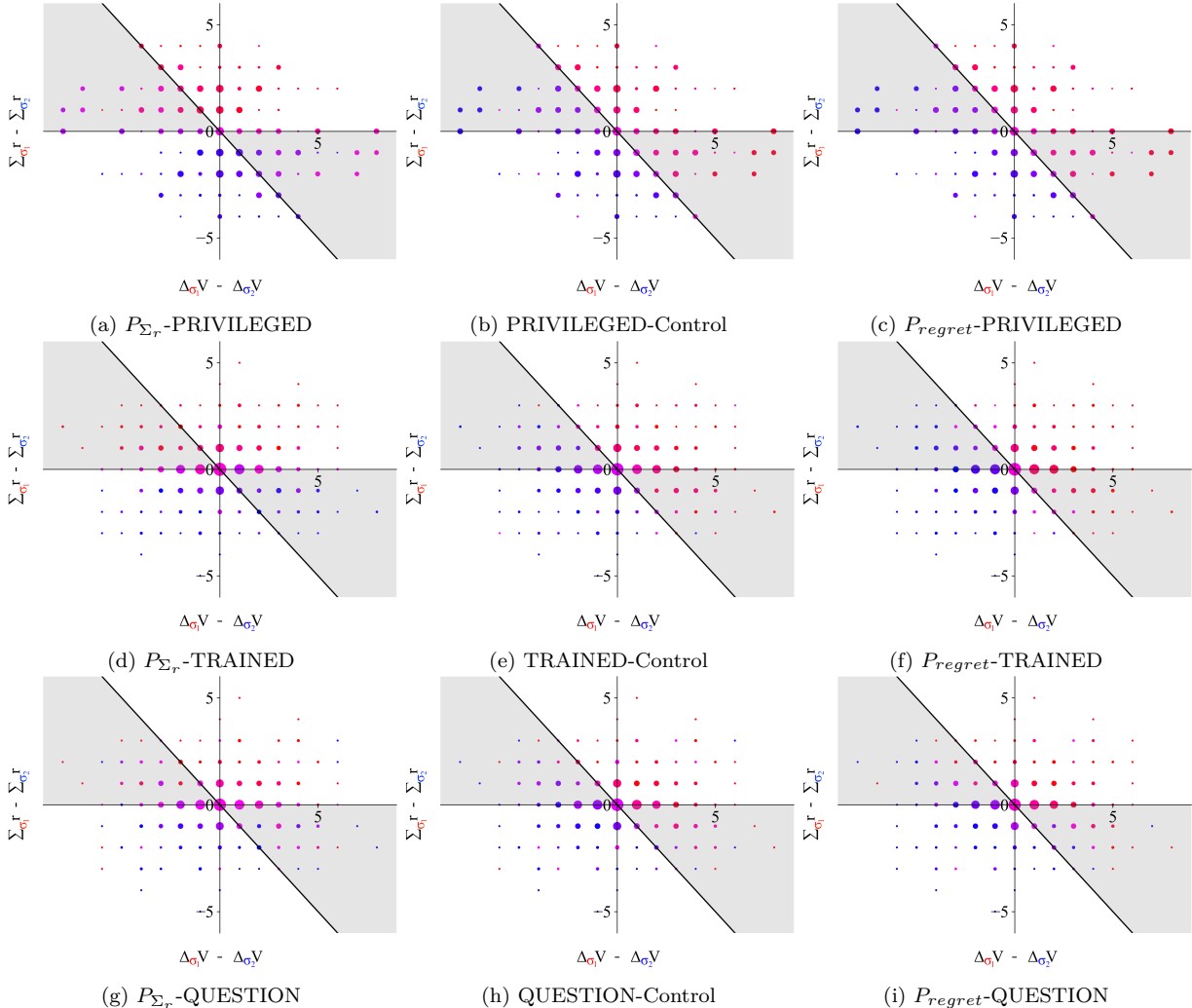

Figure 25: Proportions of subjects preferring each segment in a pair, plotted by the difference in segments' change in state values (x-axis) and partial returns (y-axis). The areas of the circles are proportional to the number of segment pairs at that point. The proportionality is consistent across across all plots for the TRAINED and QUESTION experiments, and separately for the PRIVILEGED experiments. The diagonal line indicates points of indifference for $P_{regret}$, while indifference points for $P_{\Sigma_r}$ are on the x-axis. The shaded gray area highlights where the partial return and regret preference models disagree, each preferring a different segment. To visually assess which preference model better fits the data: if subjects used the partial return preference model to generate preferences, the color gradient would be orthogonal to the x-axis. Conversely, if they followed the regret preference model, the gradient would be orthogonal to the diagonal line, as regret here is $x + y$.

# I  Likelihood of preference dataset

When we evaluate the likelihood of a preference dataset given a preference model (under the assumption that it follows a Boltzmann distribution), we seek to evaluate which class of preference model can better express the human data, given equivalent versions of the ground-truth reward function that was taught to human subjects prior to preference elicitation. Specifically, we note that reward functions that differ only by a constant scaling factor are equivalent under most definitions—including how they order policies given a start state distribution and, by consequence, their sets of optimal policies—and different scalings of the same ground-truth reward function are considered an equivalence class. Concretely, to evaluate the likelihood of a dataset given a preference model class and this equivalence class that includes the ground-truth reward function, we use the highest likelihood across a predefined list of positive scaling parameters, each of which

multiplies the output of the ground-truth reward function. This scaling parameter can also be seen as scaling the difference in the two segment statistics; mathematically, it has the same effect as using a Boltzmann temperature parameter. The predefined list of scaling parameters was chosen to cover the space in which these preference models have relatively high likelihoods. Alternatively, this scaling parameter could be learned via gradient descent and tested on a heldout set, like in k-fold cross validation, but we decided against this approach out of concerns that learning a scaling parameter is not representative of actually learning a reward function; the reward function has unknown parameters beyond its scale.

For each preference dataset resulting from each experimental condition, we evaluate how well $P_{regret}$ and $P_{\Sigma_r}$ predict the dataset. We explore a range of possible reward scaling parameters for $P_{regret}$ and $P_{\Sigma_r}$, computing the mean cross-entropy loss for each parameter and model over the dataset. The reward scaling parameters were selected to be exponentially spaced between approximately 1 and $-1$. The $n$-th reward scaling parameter is given by $p_n = ar^{n-1}$. We used 25 reward scaling parameters: the first 12 were generated with $a = 0.01$ and $r = 1.236$, the next 12 with $a = -0.01$ and $r = 1.236$, and the final parameter was set to 0.

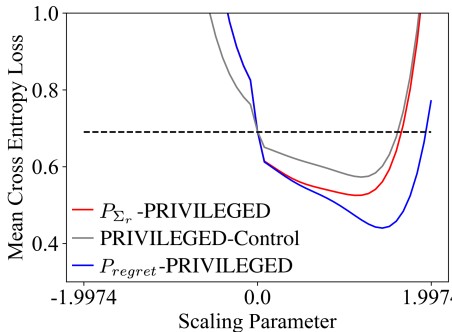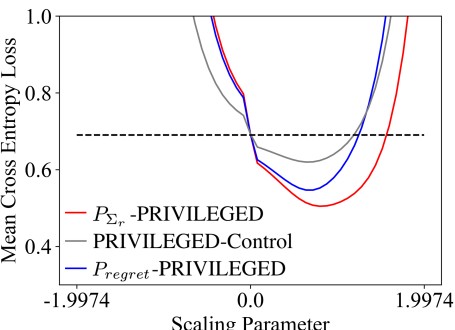

Figure 26: The mean cross entropy loss over each preference dataset resulting from the PRIVILEGED experiment (see Section 5.1) for all scaling parameters given the regret preference model (Left) and the partial return preference model (Right). If the loss is **lower** for the dataset of preferences influenced towards the target preference model than the control condition's dataset at a specific scaling parameter, it means the former is better predicted by—and more likely under—the target preference model given that scaling parameter. The regret preference model achieves the lowest mean cross-entropy loss over—and is therefore most predictive of—the $P_{regret}$-TRAINED dataset for all scaling parameters greater than 0. Similarly, the partial return preference model best predicts the $P_{\Sigma_r}$-TRAINED dataset for all scaling parameters greater than 0. This supports our hypothesis that showing subjects PRIVILEGED information about each segment's regret or partial return during preference elicitation does influence their preferences towards a specific preference model.

The plotted losses for each reward scaling parameter are illustrated in Figure 26 for the preference datasets obtained from the PRIVILEGED experiment, Figure 29 for the TRAINED experiment, Figure 30 for the TRAINED-DIFF-CONDITION experiment, Figure 27 for the QUESTION experiment, and 28 for the QUESTION-STOCHASTIC-MDP experiment. Note that when fitting the regret preference model to a preference dataset for these plots, we apply the scaling parameter to the negated regret of a segment for easier visual comparison. These plots extend the results shown in Figures 6, 7, 10, 12, 16. One incorrect conclusion to draw from those figures is that the proposed interventions simply train humans to better understand the ground-truth reward function, rather than to follow a specific preference model. We acknowledge the possibility of entangled effects relating to learning more about the ground-truth reward function rather than a specific preference model. However, we expect that in our experiments such effects are relatively minimal. Firstly, all human subjects already had a good understanding of the ground-truth reward function; we employed a comprehension test to filter out subjects who did not (see Appendix C.4). Further, in Figures 6, 7, and 12, we see that for all experiments the loss over a condition's dataset is lower under the target preference model than all other conditions datasets under the same preference model. Had one condition trivially resulted in subjects better understanding the ground-truth reward function rather than the target preference model, we would expect to see that condition's dataset induce the lowest loss under either preference model. Figures 6, 7,7, and 12 illustrate that this is not the case.

We further characterize whether there is a difference in the likelihood of the control condition's dataset and the likelihood of the dataset that arises from influencing humans towards a specific preference model. The dataset of preferences in the PRIVILEGED experiment is unpaired so we perform a Mann-Whitney U test, while the dataset of preferences in the TRAINED and QUESTION experiments are paired so we perform a Wilcoxon paired signed-rank test. See Appendix C.2 for more details on how these datasets were constructed. All statistical tests are applied between the likelihoods over each dataset that result from the best scaling parameter—meaning the scaling parameter that induces the lowest mean cross-entropy loss.

Performing the Mann-Whitney U test between $P_{\Sigma_r}$-PRIVILEGED and PRIVILEGED-Control datasets results in $U = 1418607.0$, $p < 0.01$, and between the $P_{regret}$-PRIVILEGED and PRIVILEGED-Control datasets results in $U = 1643392.5$, $p < 0.01$. In the TRAINED and QUESTION experiments, to ensure that each condition contains the same segment pairs, we removed a segment pair from all conditions if any subject selected "Can't Tell" instead of indicating a preference for that pair.

Performing the Wilcoxon test between the $P_{\Sigma_r}$-TRAINED and TRAINED-Control datasets results in $W = 6366.0$, $p < 0.01$, and between the $P_{regret}$-TRAINED and TRAINED-Control datasets results in $W = 12083.0$, $p < 0.01$. In the subsequent follow-up experiment, performing the Wilcoxon test between the $P_{\Sigma_r}$-TRAINED-DIFF-DOMAIN and TRAINED-DIFF-DOMAIN-Control datasets results in $W = 6373.0.0$, $p < 0.01$, and between the $P_{regret}$-TRAINED-DIFF-DOMAIN and TRAINED-DIFF-DOMAIN-Control datasets results in $W = 3046.5$, $p > 0.05$.

Additionally, performing the Wilcoxon test between the $P_{\Sigma_r}$-QUESTION and QUESTION-Control datasets results in $W = 7321.0$, $p < 0.05$—which we consider statistically significant—and between the $P_{regret}$-QUESTION and QUESTION-Control datasets results in $W = 2217.5$, $p = 0.685$—which we do not consider statistically significant. In the subsequent follow-up experiment, performing the Wilcoxon test between the $P_{regret}$-QUESTION-STOCHASTIC-MDP and QUESTION-STOCHASTIC-MDP-Control datasets results in $W = 4674.0$, $p = 0.412$. Performing the Wilcoxon test between the $P_{\Delta-\text{expected-return}}$-QUESTION-STOCHASTIC-MDP and QUESTION-STOCHASTIC-MDP-Control datasets results in $W = 3452.5$, $p = 0.876$. We do not consider either of these results statistically significant.

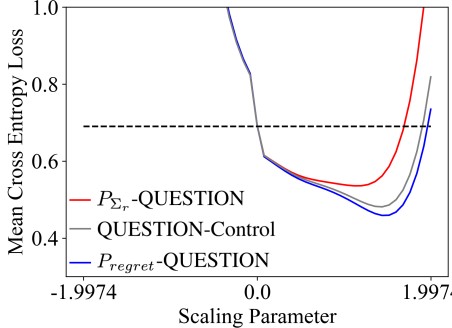 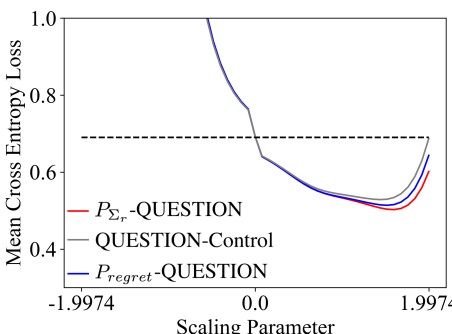

Figure 27: The mean cross entropy loss over each preference dataset resulting from the QUESTION experiment (see Section 5.3) for all scaling parameters given the regret preference model (Left) and the partial return preference model (Right). See Figure 26 for details on how to interpret this graph. The regret preference model achieves the lowest mean cross-entropy loss over the $P_{regret}$-QUESTION dataset, closely followed by the QUESTION-Control dataset, for all scaling parameters greater than 0. The partial return preference model best predicts the $P_{\Sigma_r}$-QUESTION condition's dataset, closely followed by the $P_{regret}$-QUESTION dataset and then the QUESTION-Control dataset for all scaling parameters greater than 0. These results suggest that changing the preference elicitation QUESTION to influence preferences towards the regret preference model may not be effective, though the loss over the QUESTION-Control dataset given the regret preference model is already relatively low. Modifying the QUESTION to guide preferences towards the partial return preference model has a moderate effect on the datasets conformity to the preference model.

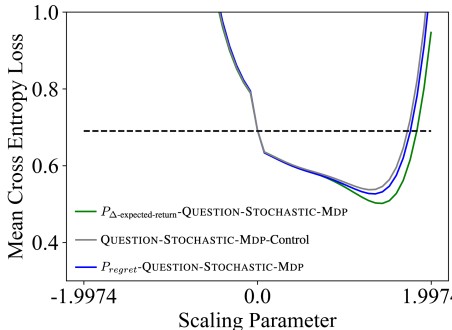 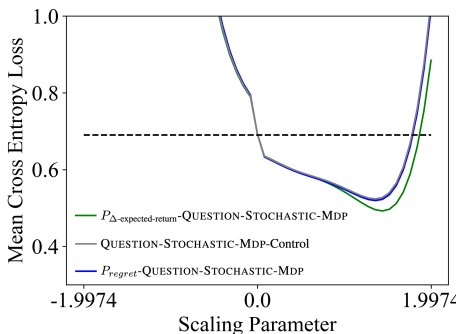

Figure 28: The mean cross entropy loss over each preference dataset resulting from the QUESTION-STOCHASTIC-MDP experiment (see Section 5.3) for all scaling parameters given the regret preference model (Left) and the change-in-expected-return preference model (Right). See Figure 26 for details on how to interpret this graph. The regret preference model achieves the lowest mean cross-entropy loss over the $P_{\Delta\text{-expected-return}}$-QUESTION-STOCHASTIC-MDP dataset, closely followed by the $P_{regret}$-QUESTION-STOCHASTIC-MDP dataset, for all scaling parameters greater than 0. The change-in-expected-return preference model best predicts the $P_{\Delta\text{-expected-return}}$-QUESTION-STOCHASTIC-MDP condition's dataset, closely followed by the $P_{regret}$-QUESTION-STOCHASTIC-MDP dataset and then the QUESTION-Control dataset for all scaling parameters greater than 0.

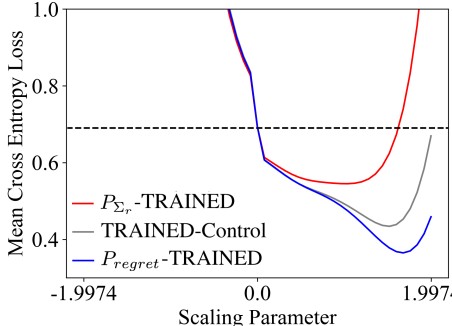 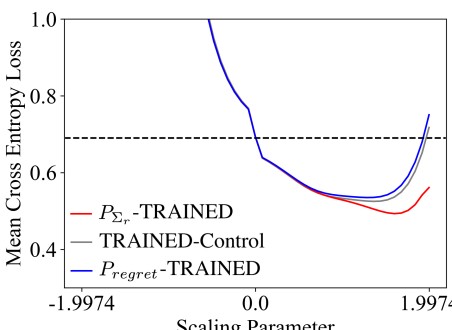

Figure 29: The mean cross entropy loss over each preference dataset resulting from the TRAINED experiment (see Section 5.2) for all scaling parameters given the regret preference model (Left) and the partial return preference model (Right). See Figure 26 for details on how to interpret this graph. For scaling parameters greater than 0, the regret preference model achieves the lowest mean cross-entropy loss over—and is therefore most predictive of—the $P_{regret}$-TRAINED dataset, followed by the TRAINED-Control dataset, and finally the $P_{\Sigma_r}$-TRAINED dataset. Similarly, the partial return preference model best predicts the $P_{\Sigma_r}$-TRAINED dataset followed by the $P_{regret}$-TRAINED and TRAINED-Control condition's datasets. This supports our hypothesis that teaching subjects about a specific preference model does influence their preferences towards that model.

## J  QUESTION-STOCHASTIC-MDP experiment: mixed preference model likelihood results

In Figure 16 we observe that changing the preference elicitation question in favor of a preference model results in a dataset that is more likely under that preference model than the control condition's dataset, but the effect size is relatively small and not statistically significant when performing a Wilcoxon test (see Appendix I for details). We do observe however, that changing the preference elicitation question in favor of the change-in-expected-return preference model results in a dataset that is significantly more likely under *the regret preference model* than the control condition's dataset.

Performing the Wilcoxon test between the likelihood of the $P_{\Delta-\text{expected-return}}$-QUESTION-STOCHASTIC-MDP and QUESTION-STOCHASTIC-MDP-Control datasets under the regret preference model results in

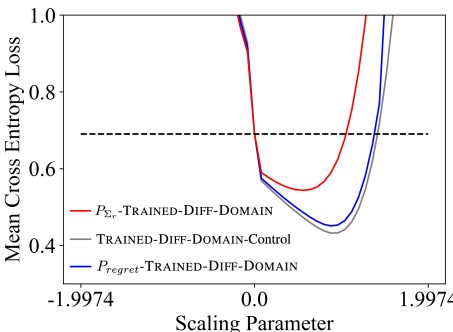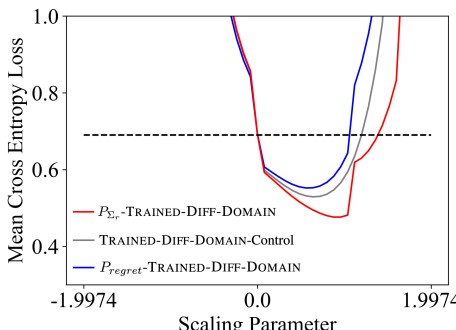

Figure 30: The mean cross entropy loss over each preference dataset resulting from the TRAINED-DIFF-DOMAIN experiment (see Section 5.2) for all scaling parameters given the regret preference model (Left) and the partial return preference model (Right). See Figure 26 for details on how to interpret this graph. For scaling parameters greater than 0, the regret preference model achieves the lowest mean cross-entropy loss over—and is therefore most predictive of—the TRAINED-DIFF-DOMAIN-Control dataset, followed by the $P_{regret}$-TRAINED-DIFF-DOMAIN dataset, and finally the $P_{\Sigma_r}$-TRAINED-DIFF-DOMAIN dataset. The partial return preference model best predicts the $P_{\Sigma_r}$-TRAINED-DIFF-DOMAIN dataset followed by the TRAINED-DIFF-DOMAIN-Control and $P_{regret}$-TRAINED-DIFF-DOMAIN condition's datasets.

$W = 26671.0$, $p < 0.01$. This indicates that changing the preference elicitation question did significantly influence human preferences towards the unintended preference model.

## K  Accuracy over preference dataset

Computing the likelihood of a preference dataset given a preference model is an informative measure of how well that preference model describes the dataset. But, computing that likelihood also requires preference model $P$—defined in Equation 2 for partial return and Equation 4 for regret—which rests on the assumption that humans are Boltzmann-rational as instantiated via the logistic function. Therefore, to circumvent this assumption, we also compute the accuracy of the *noiseless* version of a given preference model over each dataset. These results are detailed below for all experiments.

We test the significance of these results using the Fisher's exact test (Upton, 1992). When executing the Fisher's exact test, for each condition in each experiment we construct a 2x2 contingency table where the first row is the number of preferences the noiseless target preference model classified correctly, the second row is the number of preferences the noiseless target preference model classified incorrectly, the first column is the dataset where subjects are influenced towards the target preference model, and the second column is the control condition's dataset.

**PRIVILEGED Experiment**     Table 5 shows that, consistent with the results in Section 5.1, the noiseless version of the target preference preference model achieves higher accuracy on the preference dataset influenced toward the target model compared to the control condition dataset.

We conduct a Fisher exact test (Upton, 1992) to determine whether there is a significant difference in the proportion of preferences that the noiseless target preference model correctly classifies between the influenced preference dataset and the control condition dataset. We find a p-value of less than 0.001 when comparing the $P_{\Sigma_r}$-PRIVILEGED dataset to the PRIVILEGED-Control dataset, as well as when comparing the $P_{regret}$-PRIVILEGED dataset to the PRIVILEGED-Control dataset, indicating statistical significance.

**TRAINED Experiment**     Table 6 presents the accuracy of the noiseless target preference model when predicting the $P_{\Sigma_r}$-TRAINED and $P_{regret}$-TRAINED datasets compared to the TRAINED-Control dataset. The accuracy over both the $P_{\Sigma_r}$-TRAINED and $P_{regret}$-TRAINED datasets is notably higher than the accuracy over the TRAINED-Control dataset given the respective preference model, supporting the results in Section 5.2.

Table 6: The accuracy of each condition's preference dataset from the TRAINED experiment with respect to the *noiseless* version of the target preference model. Higher is better. See Table 5 for more details on how to interpret this table.

| Condition | Noiseless-$P_{\Sigma_r}$ Accuracy | Noiseless-$P_{regret}$ Accuracy |
|---|---|---|
| Control Condition | 53.7% | 68.8% |
| Influenced Towards Target Model | 75.5% | 75.4% |

We conduct the Fisher exact test over the proportion of preferences predicted correctly by the noiseless target preference model; we find a p-value of 0.0012 when comparing the $P_{\Sigma_r}$-TRAINED dataset to the TRAINED-Control dataset, and of 0.0025 when comparing the $P_{regret}$-TRAINED dataset to the TRAINED-Control dataset, indicating statistical significance.

Table 5: The accuracy of each condition's preference dataset from the PRIVILEGED experiment with respect to the *noiseless* version of the target preference model. If the accuracy is **higher** for the dataset of preferences influenced towards the target preference model than the control condition's dataset, it means the former is better predicted by the noiseless version of the target preference model.

| Condition | Noiseless-$P_{\Sigma_r}$ Accuracy | Noiseless-$P_{regret}$ Accuracy |
|---|---|---|
| Control Condition | 48.6% | 55.9% |
| Influenced Towards Target Model | 75.3% | 75.8% |

**TRAINED-DIFF-DOMAIN Experiment** Table 7 presents the accuracy of the noiseless target preference model when predicting the $P_{\Sigma_r}$-TRAINED-DIFF-DOMAIN and $P_{regret}$-TRAINED-DIFF-DOMAIN datasets compared to the TRAINED-DIFF-DOMAIN-Control dataset. The accuracy over the $P_{\Sigma_r}$-TRAINED-DIFF-DOMAIN dataset is significantly higher than the accuracy over the TRAINED-DIFF-DOMAIN-Control dataset given the partial return preference model, while the accuracy for the $P_{regret}$-TRAINED-DIFF-DOMAIN dataset roughly matches that of the TRAINED-DIFF-DOMAIN-Control dataset under the regret preference model. These results match the results in Section 5.2.

Conducting a Fisher exact test results in a p-value of 0.049 when comparing the $P_{\Sigma_r}$-TRAINED-DIFF-DOMAIN dataset to the TRAINED-DIFF-DOMAIN-Control dataset, and of 0.942 when comparing the $P_{regret}$-TRAINED-DIFF-DOMAIN dataset to the TRAINED-DIFF-DOMAIN-Control dataset.

**QUESTION Experiment** As shown in Table 8, the accuracy of the noiseless partial return preference model is higher over the $P_{\Sigma_r}$-QUESTION dataset than the QUESTION-Control dataset. This indicates that changing the preference elicitation instruction to influence preferences towards partial return results in a preference dataset that is better predicted by partial return. The $P_{regret}$-QUESTION dataset, on the other hand, is not better predicted by regret than the control condition.

We conduct Fisher's exact test and find a p-value of 0.0157 when comparing the proportion of preferences predicted correctly by the noiseless partial return preference model for the $P_{\Sigma_r}$-TRAINED dataset versus the TRAINED-Control dataset, indicating statistical significance. We do not find a statistically-significant p-value when comparing the $P_{regret}$-TRAINED dataset to the TRAINED-Control dataset ($p = 0.7144$).

**QUESTION-STOCHASTIC-MDP Experiment** Table 9 shows that the accuracy of the $P_{\Delta\text{-expected-return}}$-QUESTION-STOCHASTIC-MDP dataset under the change-in-expected-return preference model is greater than the accuracy of the control condition's dataset, but the accuracy of the control condition's dataset is higher than the $P_{regret}$-QUESTION-STOCHASTIC-MDP dataset under the regret preference model. Performing a Fisher's exact test does not result in statistical significance for either.

Table 7: The accuracy of each condition's preference dataset from the TRAINED-DIFF-CONDITION experiment with respect to the *noiseless* version of the target preference model. Higher is better. See Table 5 for more details on how to interpret this table.

| Condition | Noiseless-$P_{\Sigma_r}$ Accuracy | Noiseless-$P_{regret}$ Accuracy |
|---|---|---|
| Control Condition | 60.0% | 72.92% |
| Influenced Towards Target Model | 86.29% | 72.58% |

Table 8: The accuracy of each condition's preference dataset from the QUESTION experiment with respect to the *noiseless* version of the target preference model. Higher is better. See Table 5 for more details on how to interpret this table.

| Condition | Noiseless-$P_{\Sigma_r}$ Accuracy | Noiseless-$P_{regret}$ Accuracy |
|---|---|---|
| Control Condition | 54.8% | 69.9% |
| Influenced Towards Target Model | 65.2% | 68.7% |

Table 9: The accuracy of each condition's preference dataset from the QUESTION-STOCHASTIC-MDP experiment with respect to the *noiseless* version of the target preference model. Higher is better. See Table 5 for more details on how to interpret this table.

| Condition | Noiseless-$P_{\Delta\text{-expected-return}}$ Accuracy | Noiseless-$P_{regret}$ Accuracy |
|---|---|---|
| Control Condition | 67.8% | 65.9% |
| Influenced Towards Target Model | 69.4% | 61.2% |

## L  Learning reward functions from preferences

### L.1  Design pattern for learning a reward function from preferences

We follow the general procedure for learning a reward function from a dataset of preferences depicted in Figure 31. This procedure is executed for all preference datasets in each experiment, which all share the same ground-truth reward function $r$.

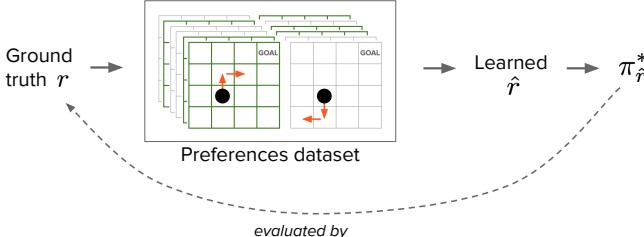

Figure 31: An outline of the general procedure for learning a reward function from a preference dataset and then evaluating that reward function. The generic gridworld shown is for illustrative purposes only. Figure provided by Knox et al. (2024).

### L.2  Additional details for learning a reward function from preferences

**Doubling the preference dataset by reversing preference samples**    When learning a reward function from a preference dataset, we double the amount of data by duplicating each preference sample and then flipping the preference label and segment pair ordering. This provides more training data and avoids learning any segment ordering effects.

**The reward representation**    The ground-truth reward function $r$ is assumed to be a linear combination of weights and features. Any reward function learned from a preference dataset $\hat{r}$ takes the same form. This linearity assumption enables us to use the tractable algorithm for learning a reward function with $P_{regret}$ proposed by Knox et al. (2022).

**Discounting during value iteration**    The delivery domain is an episodic environment but a policy derived from a poorly learned reward function can endlessly avoid terminal states, resulting in a return of negative infinity. Therefore during value iteration and when computing a policy's mean return with respect to $r$, we apply a discount factor of $\gamma = 0.999$. We chose this high discount factor to avoid returns of negative infinity while having a negligible effect on the returns of high-performing policies and still allowing value iteration to converge within a reasonable time.

**Early stopping when learning with $P_{regret}$**    Knox et al. (2022) found that when learning a reward function using $P_{regret}$, the training loss tended to fluctuate cyclically. To handle this, they use the $\hat{r}$ that achieved the lowest loss during training instead of the final $\hat{r}$. We follow the same procedure.

### L.3  Hyperparameters for learning a reward function from preferences

The following hyperparameters were used by Knox et al. (2024) and across all our experiments. See Knox et al. (2024) for more details on how they were chosen.

**Reward learning with the partial return preference model**
learning rate: 2; number of training epochs: $30,000$; and optimizer: Adam (with $\beta_1 = 0.9$ and $\beta_2 = 0.999$, and eps$= 1e - 08$).

**Reward learning with the regret preference model**
learning rate: 0.5; number of training epochs: $5,000$; optimizer: Adam (with $\beta_1 = 0.9$, $\beta_2 = 0.999$, and eps$=1e - 08$); and softmax temperature: 0.001.

Additionally, learning with $P_{regret}$ following the algorithm proposed by Knox et al. (2024) requires a set of successor features from candidate policies which are used to approximate $V_{\hat{r}}^*(.)$, a component of the regret

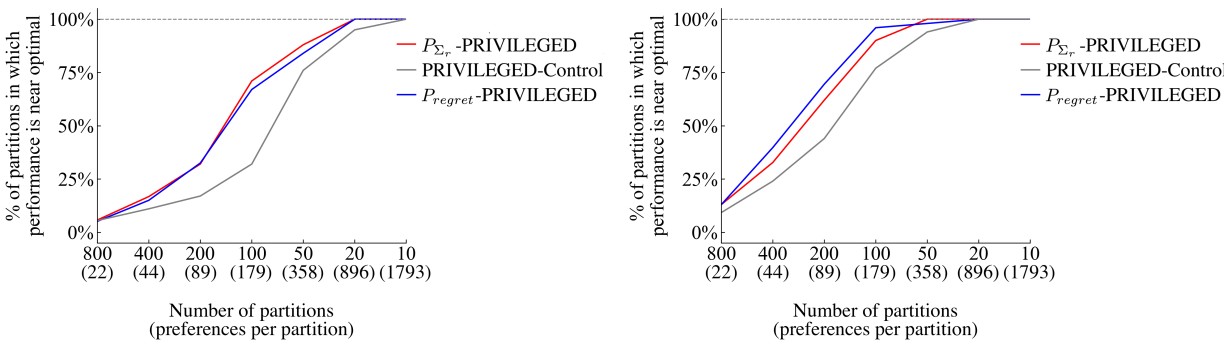

Figure 32: Learning a reward function with the partial return preference model (Left) and regret preference model (Right) from the preferences collected in the PRIVILEGED experiment. See Figure 5 for more details on how this figure was generated.

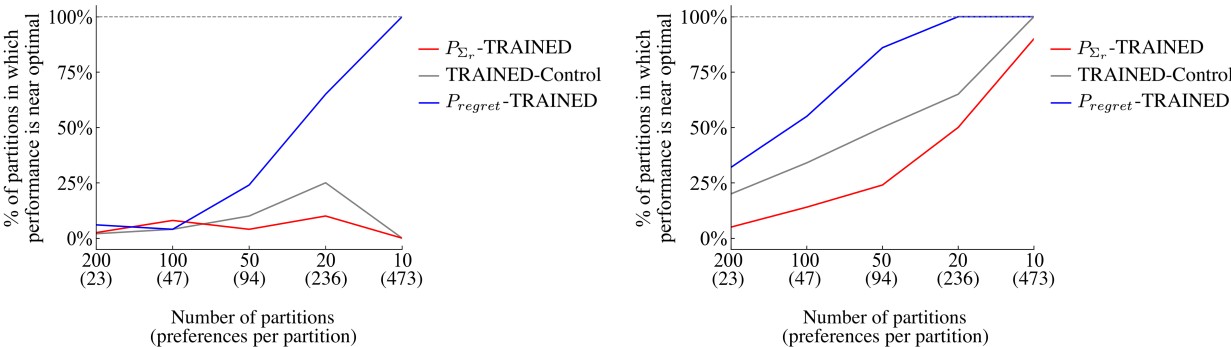

Figure 33: Learning a reward function with the partial return preference model (Left) and regret preference model (Right) from the preferences collected in the TRAINED experiment. See Figure 5 for more details on how this figure was generated.

preference model. Because we use the same delivery ask as Knox et al. (2024), we use the set of successor features that they generate.

When generating Figures 5, 8, 13, 38, and 39, we use random seeds $1 - 10$. The human preference datasets contain varying numbers of preferences (see Table 3). For all datasets within an experiment, we randomly subsample preferences to match the size of the smallest dataset—1793 preferences for the PRIVILEGED experiment datasets, 473 preferences for the TRAINED experiment datasets and 434 preferences for the QUESTION experiment datasets. This allows for easier comparison when partitioning the resulting datasets.

## L.4   Performance when learning from different preference models

Figure 32 complements Figure 5, Figure 33 complements Figure 8, Figure 34 complements Figure 11, Figure 35 complements Figure 13, and Figure 36 complements Figure 15. For each experiment and condition, these plots show the performance of learning a reward function with either preference model used in that experiment, rather than only the target preference model.

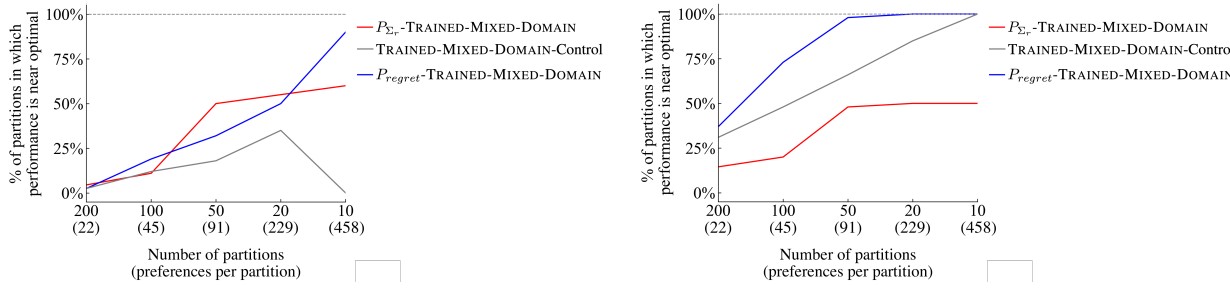

Figure 34: Learning a reward function with the partial return preference model (Left) and regret preference model (Right) from the preferences collected in the TRAINED-DIFF-DOMAIN experiment. See Figure 5 for more details on how this figure was generated.

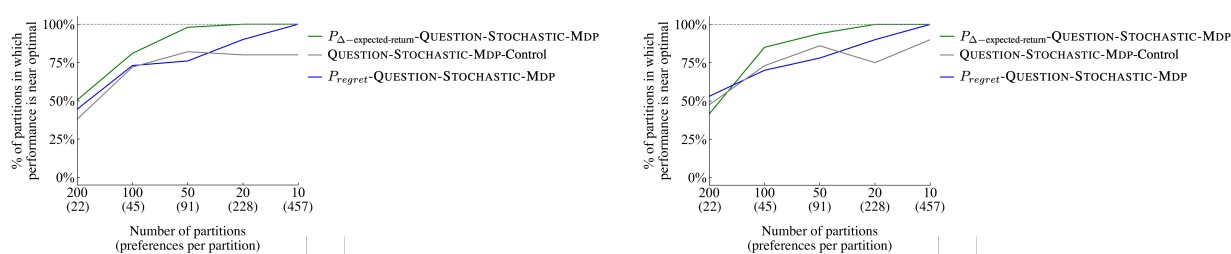

Figure 36: Learning a reward function with the change-in-expected-return preference model (Left) and regret preference model (Right) from the preferences collected in the QUESTION-STOCHASTIC-MDP experiment. See Figure 5 for more details on how this figure was generated.

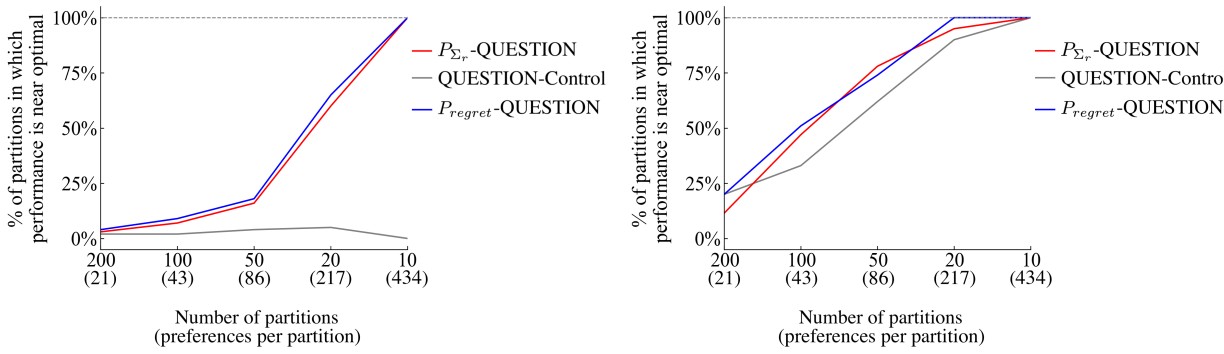

Figure 35: Learning a reward function with the partial return preference model (Left) and regret preference model (Right) from the preferences collected in the QUESTION experiment. See Figure 5 for more details on how this figure was generated.

## L.5   Performance compared to a uniformly random policy

Figure 37 complements Figure 5, Figure 38 complements Figure 8, and Figure 39 complements Figure 13, showing the percentage of partitions where the learned reward functions results in better performance than a policy that selects actions uniformly. In general, for each partition size, ranking each preference dataset by the percentage of better-than-random performance induced by the learned reward functions produces the same order as when using near-optimal performance.

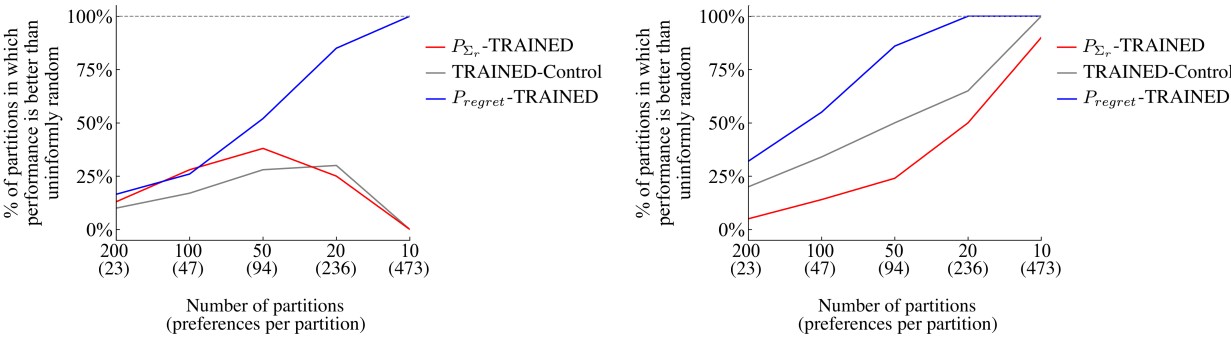

Figure 38: Learning a reward function with the partial return preference model (Left) and regret preference model (Right) from the preferences collected in the TRAINED experiment. This figure complements Figure 8.

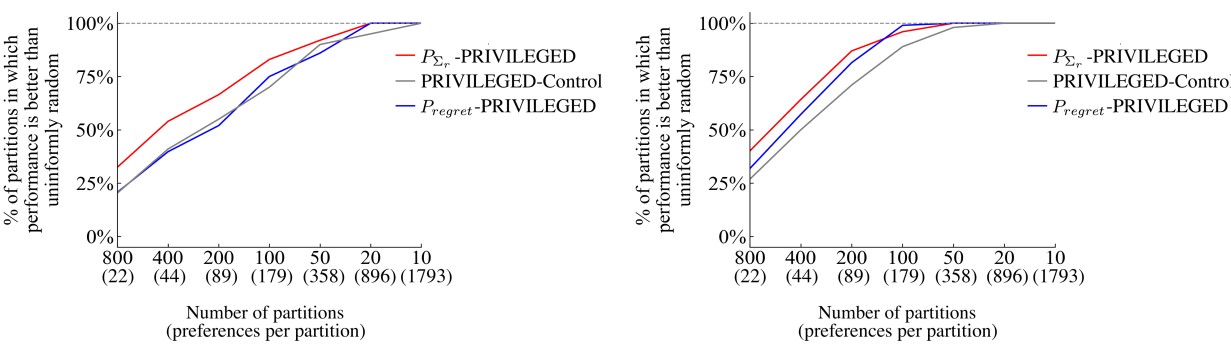

Figure 37: Learning a reward function with the partial return preference model (Left) and regret preference model (Right) from the preferences collected in the PRIVILEGED experiment. This figure complements Figure 5.

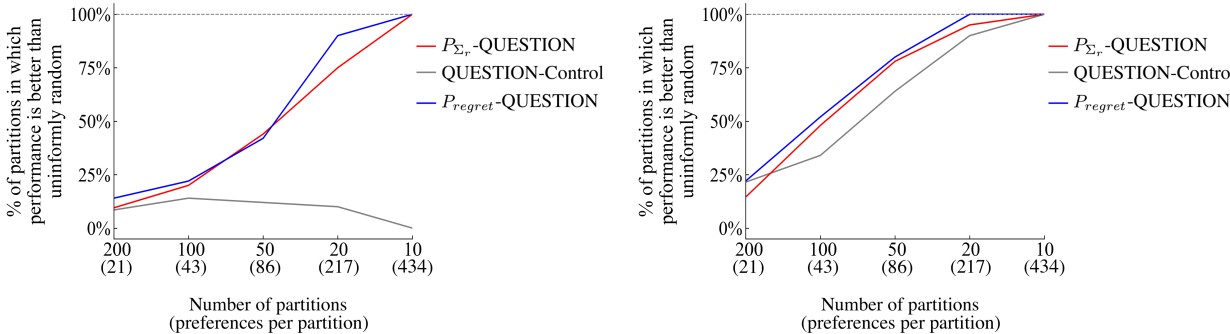

Figure 39: Learning a reward function with the partial return preference model (Left) and regret preference model (Right) from the preferences collected in the QUESTION experiment. This figure complements Figure 13.

### L.6 Addressing the partial return preference model's identifiablity issues

Learning with the partial return preference model from the $P_{\Sigma_r}$-TRAINED and TRAINED-Control datasets often fails to recover a reward function that induces near-optimal performance (see Figure 8). Knox et al. (2022) demonstrated that, in this grid-world domain, learning with the partial return preference model requires preference labels over pairs of trajectory segments in which one segment terminates earlier than the other at both the positive and negative terminal states (i.e., the inverted teardrop and sheep in Figure 3). Such segment pairs help mitigate the identifiability issue of the partial return preference model related to a constant shift in the reward function. To address this issue, Knox et al. (2022) manually selected types of segment pairs for preference labeling. We follow their segment pair selection methodology for the PRIVILEGED experiment, but for the TRAINED and QUESTION experiments, trajectory segments were constructed by randomly

sampling actions to emulate a more realistic preference elicitation procedure. Consequently the partial return preference model may not recover the ground-truth reward function from the resulting preference datasets, which we hypothesize as explaining the partial return preference model's poor performance when learning from the $P_{\Sigma_r}$-TRAINED and TRAINED-Control datasets in Figure 8. Knox et al. (2024) showed that using preferences generated by $P_{regret}$ to learn a reward function with the partial return preference model results in a reward function that is equivalent to an optimal advantage function, which may explain why the partial return preference model recovers performant reward functions from the $P_{regret}$-TRAINED dataset. We leave an investigation into this hypothesis to future work.

To empirically test whether the absence of specific segment pairs contributed to the poor performance of the partial return preference model when learning reward functions from the $P_{\Sigma_r}$-TRAINED and TRAINED-Control datasets, we added 50 additional segment pairs to each dataset. In these additional segment pairs, one segment terminates at the positive terminal state in fewer than three time-steps while the other segment does not terminate after three time-steps. These segment pairs would appear in the right-most graph in Figure 20, and were assigned preference labels by the partial return preference model with the ground-truth reward function. Figure 40 present the results when learning reward functions using these additional *synthetic* preferences.

Learning with the regret preference model using these additional preferences (bottom row of Figure 40) induces comparable results to those in Figure 8 and Figure 38 which matches our expectations; the regret preference model does not suffer from the same identifiability issues as the partial return preference model. Learning with the partial return preference model when including the additional preferences (top row of Figure 40) results in reward functions that induce near-optimal behavior more often for all datasets across all partition sizes. Including these preferences also results in better-than-uniformly-random behavior significantly more often across all partition sizes and datasets. These results therefore support our hypothesis that the partial return preference model's poor performance when learning from the $P_{\Sigma_r}$-TRAINED and TRAINED-Control datasets is—at least in part—due to the datasets missing specific segment pairs that account for the partial return preference model's identifiability issues.

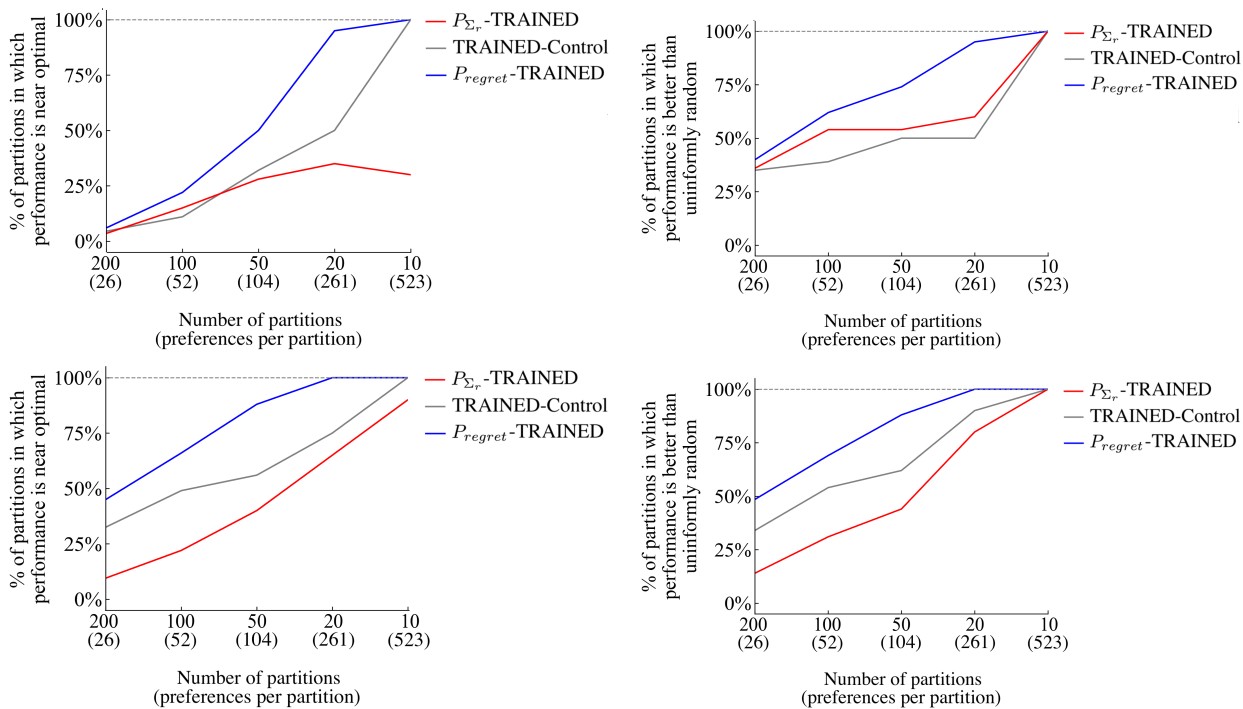

Figure 40: Learning a reward function with the partial return preference model (Top Row) and regret preference model (Bottom Row) from *synthetic* and human preferences. The TRAINED experiment's preference datasets are partitioned following the same methodology as when generating Figure 8. Each preference dataset contains 50 additional segment pairs with synthetic preference labels that aim to compensate for the identifaibility issues of the partial return preference model.

