# OpenReview forum: "Influencing Humans to Conform to Preference Models for RLHF"
_TMLR — Under review for TMLR_

### Review · Reviewer_eaNh · 2026-05-15

**Summary Of Contributions:**

This paper studies a novel perspective on Reinforcement Learning from Human Feedback (RLHF): instead of improving the preference model to better fit human behavior, the authors investigate whether humans can be influenced to express preferences that better conform to a target preference model assumed by the RLHF algorithm.

Strengths:
- The paper introduces a fresh perspective on RLHF alignment. Rather than only modifying the learning algorithm or preference model, it studies how human preference expression itself can be shaped through interface and training design.
- The authors evaluate several intervention types across deterministic and stochastic environments, including cross-domain transfer experiments.
- In my view, one of the most important aspects of the paper is the carefully collected human preference dataset under different intervention settings. The dataset enables controlled analysis of how elicitation interfaces alter preference distributions and could support future work on RLHF preference modeling.

Weakness:
- The conclusions rely substantially on the constructed grid-world datasets and the specific elicitation interfaces. It is unclear whether the observed effects generalize to realistic RLHF settings such as LLM alignment or robotics.
- The TRAINED intervention appears cognitively expensive, especially for regret-based reasoning, which may limit scalability in realistic RLHF pipelines.

**Audience:**

Yes

**Audience Explanation:**

The main conceptual contribution, that preference elicitation interfaces themselves are part of the alignment pipeline, is timely and relevant to current discussions around RLHF reliability and human feedback quality.

**Broader Impact Concerns:**

I may think the paper would raise ethical concerns because intentionally steering annotators toward predefined preference models may manipulate human judgments, reduce authentic feedback, and introduce normative biases into RLHF systems. Although the authors present the approach as a way to improve alignment between humans and algorithms, similar techniques could also be misused to bias annotators toward organizational goals, suppress diverse opinions, or artificially homogenize preferences.

**Claims And Evidence:**

Yes

**Claims Explanation:**

The paper provides reasonably convincing empirical evidence for its central claim that human preference expression can be influenced toward target preference models through interface design and training interventions.

**Requested Changes:**

- The paper should more carefully discuss the limitations of transferring findings from simple grid worlds to realistic RLHF settings such as LLMs or robotics. Currently, practical implications may appear overstated.
- The paper should better explain why some interventions improve preference conformity but not downstream reward learning. The partial-return TRAINED results especially require deeper discussion.
- The computational and cognitive costs of training annotators should be discussed more concretely.

---

> ### Author Response · Authors · 2026-06-14
>
> We thank the reviewer for their thoughtful review and in-depth breakdown of the strengths and weaknesses of our work.
>
> > The conclusions rely substantially on the constructed grid-world datasets and the specific elicitation interfaces. It is unclear whether the observed effects generalize to realistic RLHF settings such as LLM alignment or robotics.
>
> We note first that the QUESTION experiment is directly applicable to settings such as LLM alignment and robotics, since it involves only changing the preference elicitation question and no domain-specific machinery. More broadly, we use simple gridworld domains—in line with the prior work of Knox et al. (2022)—so that we can elicit preferences from people without domain knowledge while retaining a clear method for evaluating the learned reward functions. We agree that extending this work to more complex and realistic domains is a valuable direction.
>
> > The TRAINED intervention appears cognitively expensive, especially for regret-based reasoning, which may limit scalability in realistic RLHF pipelines.
>
> We agree with the reviewer that the TRAINED intervention is cognitively demanding, particularly for regret-based reasoning. We would note, however, that it is considerably less demanding and significantly effective for the partial return model, suggesting it may still be useful in certain settings. We would also observe that RLHF practitioners may find a more cognitively expensive elicitation procedure an acceptable trade-off when it yields preference datasets that—as we show—induce more aligned reward functions. Finally, we do highlight that future work can explore how to reduce the cognitive load of learning to follow the regret preference model: since subjects already tend toward regret intuitively in our control conditions, training that targets fast, intuitive (System 1) reasoning rather than explicit computation may achieve comparable conformance at lower cognitive cost.
>
> > The paper should more carefully discuss the limitations of transferring findings from simple grid worlds to realistic RLHF settings such as LLMs or robotics. Currently, practical implications may appear overstated.
>
> We thank the reviewer for this suggestion. In our concluding section, we note that future research should explore similar interventions in more complex domains such as robotics and embodied agents, and throughout the paper we are explicit that all experiments are conducted in a gridworld setting—both in the text and through the numerous figures illustrating it. We are careful not to claim that our interventions will transfer to other settings; rather, we frame such transfer as an open and interesting direction for future work. For these reasons, we believe the paper does not overstate its practical implications, and we would prefer to keep the current discussion largely as is. That said, if the reviewer feels a particular passage reads as overstated, we would be glad to revise it to make the scope of our claims even clearer.
>
> > The paper should better explain why some interventions improve preference conformity but not downstream reward learning. The partial-return TRAINED results especially require deeper discussion.
>
> We thank the reviewer for raising this. We do discuss the partial-return TRAINED results—and specifically why improved preference conformity in that setting does not translate into improved downstream reward learning—in Appendix K.6.
> The computational and cognitive costs of training annotators should be discussed more concretely.
> We have added a section in the Appendix entitled “On the cognitive cost of the TRAINED experiment”. We note that the other interventions we propose are substantially less cognitively demanding.

---

> > ### Author Response · Authors · 2026-06-14
> >
> > >I may think the paper would raise ethical concerns because intentionally steering annotators toward predefined preference models may manipulate human judgments, reduce authentic feedback, and introduce normative biases into RLHF systems.
> >
> > We would like to clarify the aim of our interventions: each is designed to influence the preferences a human labels so that we can better recover their unobservable ground-truth reward function—the one that generated those preferences in the first place. At no point do we aim to alter the human's underlying reward function itself. In other words, our interventions are designed to increase conformance between expressed preferences and a chosen preference model, not to reshape what people actually value.
> >
> > We agree that other interventions—or misapplications of similar techniques—could influence human preferences in the harmful ways the reviewer describes. This is a real concern. We would note, however, that it is a concern shared by essentially any method for aligning AI systems with human goals: such methods can, in principle, be used to align AI systems with harmful goals as readily as beneficial ones. We believe that developing principled methods for people to specify their true preferences to AI systems is an important part of building human-aligned AI, and that declining to study this direction altogether would leave us building powerful systems without the tools to ensure they respect human values.

---

### Review · Reviewer_61N3 · 2026-05-28

**Summary Of Contributions:**

The paper considers RLHF training, in which  human raters make pairwise comparisons between outputs. The central assumption of RLHF is that these are based on some underlying reward function over the MDP. In particular, the most common assumption is that that preferences come from partial return over some unobservable rewards. This is a perhaps strong assumption, and a relatively large body of worked has considered other models of human preferences. This paper focuses on the models of partial return, regret, and change-in-expected-return (each over some unknown instantaneous reward).

The central focus of the paper is to test whether real humans can be trained or induced to answer RLHF comparison queries in a way that is more consistent with these preference models, and also whether this improves policies trained on this RLHF data. They consider several experimental settings:

1. the PRIVILEGED setting, which is unrealistic in that it simply tells the humans what their calculated returns/regrets/etc. are. Here, there is a very significant improvement, and training on the human-generated preference data in the treatment condition also results in much better policies.

2. the TRAINED setting, where people are trained to explicitly calculate returns, in which the training results in a statistically significant improvement (but with more mixed results when generalizing to unseen domains)

3. the QUESTION setting, where only the question asked to the human preference raters is changed. Here, results are in all but one case in the right direction, but not always statistically significant.

Overall, the paper presents mixed evidence that humans can be induced in at least one simple domain to answer RLHF queries more consistently with various reward models. A key limitation is that all experiments took place only in very simple gridworld domains. As the authors suggest, future research considering more complex domains would be of interest.

**Audience:**

Yes

**Audience Explanation:**

RLHF is of interest to many machine learning researchers, so this would certainly be of interest to at least some of them.

**Broader Impact Concerns:**

There are no ethical concerns with this work. The authors obtained IRB approval for their human subjects research.

**Claims And Evidence:**

Yes

**Claims Explanation:**

Yes, the experiments and statistical tests are explained clearly. The authors present evidence that supports each of their hypotheses and are honest about its limitations. Clear distinctions are drawn between statistically-significant and non-statistically-significant results. I have basic knowledge of statistical testing but am not a statistician, so it's possible I missed some subtleties in how the tests are applied.

**Requested Changes:**

I don't have any requested changes.

---

> ### Author Response · Authors · 2026-06-14
>
> We thank the reviewer for their thoughtful review and their careful engagement with our work. We are glad the reviewer agrees with our characterization of the results.
>
> >Overall, the paper presents mixed evidence that humans can be induced in at least one simple domain to answer RLHF queries more consistently with various reward models.
>
> We would add one clarification: while not every intervention induces a statistically significant difference in human conformance to the chosen preference model across all preference models, each intervention does induce a statistically significant difference in how humans label preferences for at least one preference model. Moreover, each intervention improves the alignment of the learned reward function in nearly all settings. This overview is highlighted in Table 1.
>
> > A key limitation is that all experiments took place only in very simple gridworld domains. As the authors suggest, future research considering more complex domains would be of interest.
>
> We use simple gridworld domains, in line with the prior work of Knox et al. (2022), so that we can elicit preferences from people without domain knowledge while retaining a clear method for evaluating the learned reward functions. We agree that extending this work to more complex domains is an important direction, and we appreciate the reviewer highlighting it as a promising avenue for future research.

---

### Review · Reviewer_UaRs · 2026-06-10

**Summary Of Contributions:**

This paper presents and experimental study into shaping human preferences by experimental design with the target to observe bias with respect to a specific preference model. The framework used is the RLHF under three different scenarios geared towards change of human behavior. The three scenarios include showing humans the real rewards of their possible actions, training people to a specific reward model and changing the preference elicitation questions. The paper is well written and is exhaustively experimental. The appendix provides enough experimental proof for the paper be reproducible.

**Audience:**

Yes

**Audience Explanation:**

The RLHF is a relatively new are for learning human preferences as a reward policy. As such it is a very timely research. In addition the presented work looks not at the task from a standard point of view but rather at the bias that directly impacts the quality of the RLHF experiments. This study is relevant as a verification of the standard approaches.

In addition I think the results of the Question based experiments are very interesting because they provide an implicit bias not in generally accounted for during human preferences mapping.

**Claims And Evidence:**

Yes

**Claims Explanation:**

The paper follows a well described methodology and format. The main experimentation is interesting but I also feel that is not providing enough of insight. In particular, the general conclusion from all three experiments can be the sensitized people will be behaving as is the expected behavior of a sensitized individual.  While the population sampling related issues and the statistics of the data has been handled properly I wonder at the relevance of the results. In particular, given that the results are very consistent with the expectations, the question that arises is that whether or not this study was performed on the appropriate level of the manipulated information.

This means that while the maze is naturally guided by individual preferences and policy of reward will also affect the decision process, the question whether the simplicity of the task does not prevent any higher level bias interference.

**Requested Changes:**

constrastive -> contrastive
page 8; from from -> from
Figure 5 should be referenced before Figure 6.

---

> ### Author Response · Authors · 2026-06-14
>
> We thank the reviewer for their review, and for their interest in our QUESTION experiment.
>
> >The main experimentation is interesting but I also feel that is not providing enough of insight. In particular, the general conclusion from all three experiments can be the sensitized people will be behaving as is the expected behavior of a sensitized individual
>
> We thank the reviewer for engaging with our experiments. We'd like to clarify the contribution, as we believe it may be more substantive than it first appears. The general conclusion across all three experiments is that one can often—though not always—influence the preferences humans express in accordance with a specific preference model by changing the elicitation procedure and interface, and that doing so improves the alignment of the learned reward function. We respectfully suggest that this is a non-obvious and highly practical finding: to our knowledge, no prior work has explored such approaches to learning reward functions from human feedback, a field of broad interest and applicability.
>
> We'd also gently push back on the framing that the conclusion amounts to "sensitized people behave as expected." While our results frequently aligned with our hypotheses, the fact that an outcome is consistent with a hypothesis does not make it unsurprising or uninformative—particularly when the underlying question (whether and how elicitation design shapes learned rewards) has not been previously examined.
>
> >given that the results are very consistent with the expectations, the question that arises is that whether or not this study was performed on the appropriate level of the manipulated information
>
> We conducted our experiments in the same domain as prior work on how humans label preferences over trajectories for RLHF [Knox et al., 2022]; this domain is designed so that preferences can be elicited from people without domain knowledge, while still providing a clear method for evaluating the learned reward functions.
>
> We want to make sure we address the reviewer's concern fully, but we are not entirely certain we understand what is meant by "the appropriate level of the manipulated information," and we would be grateful for clarification. In case it is relevant, we note that we evaluate five distinct intervention settings that vary considerably in how much information we provide to human subjects—ranging from showing them privileged information, to training them to compute the statistics of interest, to changing only the preference elicitation question and nothing else.